# Towards Realistic Mechanisms That Incentivize Federated Participation and Contribution

## Abstract

Edge device participation in federating learning (FL) is typically studied through the lens of device-server communication (*e.g.,* device dropout) and assumes an undying desire from edge devices to participate in FL. As a result, current FL frameworks are flawed when implemented in realistic settings, with many encountering the free-rider dilemma. In a step to push FL towards realistic settings, we propose REALFM: the first federated mechanism that (1) realistically models device utility, (2) incentivizes data contribution and device participation, (3) provably removes the free-rider dilemma, and (4) relaxes assumptions on data homogeneity and data sharing. Compared to previous FL mechanisms, REALFM allows for a non-linear relationship between model accuracy and utility, which improves the utility gained by the server and participating devices. On real-world data, REALFM improves device and server utility, as well as data contribution, *by over 3 and 4 magnitudes* respectively compared to baselines.

## 1 Introduction

Federated Learning (FL) is a collaborative framework where, in the *cross-device* setting, edge devices jointly train a global model by sharing locally computed model updates with a central server. It is generally assumed within FL literature that edge devices will (i) always participate in training and (ii) use all of its local data during training. However, it is irrational for devices to participate in, and incur the costs of, federated training without receiving proper benefits back from the server (via model performance or monetary rewards). Specifically, there are two major challenges:

**(C1) Lack of Participation Incentives**. Current FL frameworks generally lack incentives to increase device participation. This leads to training with fewer devices and data to compute local updates, which can potentially reduce model accuracy. Incentivizing devices to participate in training and produce more data, especially from a server's perspective, improves model performance (further detailed in Section 3), which leads to greater utility for both the devices and server.

**(C2) Lack of Contribution Incentives: The Free-Rider Dilemma**. In realistic settings, devices determine their own optimal amount of data usage for federated contributions. As such, many FL frameworks run the risk of encountering the free-rider problem: devices do not contribute gradient updates yet reap the benefits of a well-trained collaborative model. Removing the free-rider effect in FL frameworks is critical because it improves performance of trained models (Wang et al., 2023; 2022) and reduces security risks (Fraboni et al., 2021; Lin et al., 2019; Wang, 2022) for devices.

To address these challenges, a handful of recent FL literature (Karimireddy et al., 2022; Zhan et al., 2020a;b; 2021) consider device **utility**: the net benefit received by a device for participating in federated training. Every rational device $i$ aims to maximize its utility $u_i$. Consequently, utility is the guiding factor in whether rational devices participate in federated training. Devices will only participate if the utility $u_i^r$ gained, via its rewards $(a_i^r, R_i)$, outstrips the maximum utility gained from local training $u_i^o$. In a step towards more realistic FL, the referenced works (Karimireddy et al., 2022;

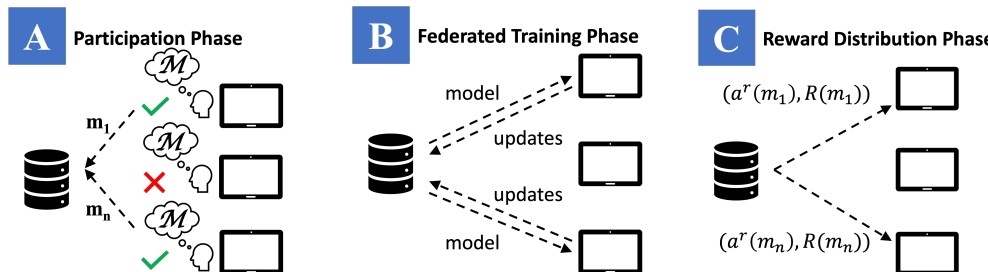

Figure 1: **Federated Mechanism Diagram. (A) Decision Phase for Device Participation**. Devices decide whether they want to participate in the mechanism. If so, the quantity of data points used by each agent $m_i$ is sent to the server (no data is shared). We note that rational devices would participate in REALFM; the utility gained by participating is *never* less than what agents attain locally. **(B) Federated Training Phase**. Devices upload their updates and receive feedback from the server in an iterative manner. **(C) Accuracy & Monetary Reward Distribution Phase.** Upon completion of federated training, the server distributes both accuracy $a^r(m_i)$ and monetary $R(m_i)$ rewards to device $i$. These rewards, the crux of REALFM, incentivize device participation and data contribution.

Zhan et al., 2020a;b; 2021) design mechanisms, or systems, that maximize device utility, and provide greater utility than local training, when more data is contributed.

While previous FL works incorporate utility, they require **unrealistic assumptions** such as disallowing **(i)** devices having utility that depends non-linearly on their model accuracy, **(ii)** heterogeneous data distributions across devices, **(iii)** truly federated (non-data sharing) methods, and **(iv)** modeling of central server utility. In our paper, we take a leap towards realistic federated systems by relaxing all 4 assumptions above while simultaneously solving the key issues in **C1** and **C2**.

**Summary of Contributions**. We propose REALFM: a federated mechanism $\mathcal{M}$ (*i.e.,* system) that a server, in a FL setup, implements to eliminate **(C1)** and **(C2)** when rational devices participate. REALFM is *Individually Rational* (IR): participating devices and the server provably receive greater utility than training alone $u_i^r \geq u_i^o$. The goal of REALFM is to *design a reward protocol, with model-accuracy $a^r$ and monetary $R$ rewards, such that rational devices choose to participate and contribute more data.* By increasing device participation and data contribution, a server trains a higher-performing model and subsequently attains greater utility. REALFM is a mechanism that,

- eliminates **(C1)** and **(C2)**, i.e., provably eliminates the free-rider effect by incentivizing devices to participate and use more data during the federated training process than they would on their own,
- allows more **realistic settings**, including: **(1)** a non-linear relationship between accuracy and utility, **(2)** data heterogeneity, **(3)** no data sharing, and **(4)** modeling of central server utility.
- produces state-of-the-art results towards improving utility (for both the server and devices), data contribution, and final model accuracy on real-world datasets.

## 2 RELATED WORKS

**Federated Mechanisms.** Previous literature (Chen et al., 2020; Zhan et al., 2020a;b; 2021) have proposed mechanisms to solve **(C1)** and incentivize devices to participate in FL training. However, these works fail to address **(C2)**, the free-rider problem, and have unrealistic device utilities. In Chen et al. (2020), data sharing is allowed, which is prohibited in the FL setting due to privacy concerns. In Zhan et al. (2020a;b; 2021), device utility incorporates a predetermined reward for participation in federated training without specification on how this amount is set by the server. This could be unrealistic, since rewards should be dynamic and depend upon the success (resulting model accuracy) of the federated training. Setting too low of a reward impedes device participation, while too high of a reward reduces the utility gained by the central server (and risks negative utility if performance lags total reward paid out). Overall, predicting an optimal reward *prior* to training is unrealistic. In contrast, our proposed REALFM introduces a principled mechanism to set the rewards.

The recent work by Karimireddy et al. (2022) is the first to simultaneously solve **(C1)** and **(C2)**. They propose a mechanism that incentivize devices to (i) participate in training and (ii) produce more data

than on their own (data maximization). By incentivizing devices to maximize production of local data, the free-rider effect is eliminated. While this proposed mechanism is a great step forward for realistic mechanisms, pressing issues remain. First, Karimireddy et al. (2022) requires data sharing between devices and the central server. This is acceptable if portions of local data are shareable (*i.e.* no privacy risks exist for certain subsets of local data), yet it violates the key tenet of FL: privacy. Second, device utility is designed in Karimireddy et al. (2022) such that the utility improves linearly with increasing accuracy. We find this unrealistic, as devices likely find greater utility for an increase in accuracy from 98% to 99% than 48% to 49%. Finally, Karimireddy et al. (2022) assumes all local data comes from the same distribution, which is unrealistic. Our proposed REALFM addresses all issues above. Further discussion is in Appendix A.

**Contract Theory and Federated Free Riding.** Contract theory in FL aims to optimally determine the balance between device rewards and registration fees (cost of participation). In contract mechanisms, devices may sign a contract from the server specifying a task, reward, and registration fee. If agreed upon, the device signs and pays the registration fee. Each device receives the reward if it completes the task and receives nothing if it fails. Contract mechanisms have the ability to punish free riding in FL by creating negative incentive if a device does not perform a prescribed task (*i.e.,* it will lose its registration fee). The works Cong et al. (2020); Kang et al. (2019); Lim et al. (2020; 2021); Liu et al. (2022); Wang et al. (2021) propose such contract-based FL frameworks. While effective at improving model generalization accuracy and utility (Lim et al., 2020; Liu et al., 2022), these works focus on optimal reward design. Our REALFM mechanism does not require registration fees, boosting device participation, and utilizes an accuracy shaping method to provide rewards at the end of training in an optimal and more realistic approach. Furthermore, like Karimireddy et al. (2022), our mechanism incentivizes increased contributions to federated training, which is novel compared with the existing contract theory literature and mechanisms.

## 3 PROBLEM FORMULATION

Within the FL setting, $n$ devices collaboratively train the parameters $\boldsymbol{w}$ of a machine learning (ML) model. Devices compute local gradient updates on $\boldsymbol{w}$ using their own local data, with the server aggregating all local device updates to perform a single global update. Each device $i$ has its own local dataset $\mathcal{D}_i$ (able to change in size and distribution) which can be heterogeneous across devices. We define the amount of data per device $i$ as $m_i := |\mathcal{D}_i|$ and denote $\boldsymbol{m} := \{m_1, \ldots, m_n\}$. Dataset sizes are constrained by cost; each device $i$ has its own fixed marginal cost $c_i > 0$ per sample, which represents the cost of collecting and computing the gradient of an extra data point (*e.g.,* collecting $m$ samples incurs a cost of $c_i m$). We consider linear costs, as data collection and sampling costs are generally constant over time in the cross-device setting (*e.g.,* powering an IoT sensor incurs a constant cost on average) (Lu et al., 2022; Tran et al., 2019; Wu et al., 2023). Overall, each device $i$ determines the dataset size $m_i$ that best balances data costs $c_i m_i$ with model performance.

**Mechanisms**. To entice devices to participate in federated training, a central server must incentivize them. Two realistic rewards that a central server can provide are: **(i)** model accuracy and **(ii)** monetary. The interaction between the server and devices is formalized as a *mechanism* $\mathcal{M}$. When participating in the mechanism, a device $i$ performs federated updates on a global model $\boldsymbol{w}$, using $m_i$ local data points, in exchange for model accuracy $a_i^r \in \mathbb{R}_{\geq 0}$ and monetary $R_i \in \mathbb{R}_{\geq 0}$ rewards.

$$\mathcal{M}(m_1, \cdots, m_n) = ((a_1^r, R_1), \cdots, (a_n^r, R_n)). \tag{1}$$

We desire a mechanism which provides rewards in proportion to the amount of contributions $m_i$ each device $i$ makes. Without proportionality, FL methods fall prey to the free-rider dilemma, where devices are able to reap rewards without proper contribution.

**Definition 1** (Feasible Mechanism). *A feasible mechanism $\mathcal{M}$ (1) returns a non-negative reward and accuracy for each device, and (2) is bounded in its provided utility.*

**Definition 2** (Individual Rationality (IR)). *A mechanism $\mathcal{M}$ is IR if devices always receive better utility by participating than it can training by itself.*

Rational devices are willing to participate in a server's mechanism $\mathcal{M}$ if they can receive (i) realistic rewards (**feasible**) and (ii) greater utility than they can get by training alone (**IR**). Finally, we must prove that mechanisms fulfilling such qualities can reach a stable equilibrium of device contributions.

**Theorem 1** (Existence of Pure Equilibrium). *Consider a feasible mechanism $\mathcal{M}$ providing utility $[\mathcal{M}^U(m_i; \boldsymbol{m}_{-i})]_i$ to device $i$. Devices receive no utility if no data is contributed, $[\mathcal{M}^U(0; \boldsymbol{m}_{-i})]_i = 0$. Define the utility of a participating device $i$ as,*

$$u_i^r(m_i; \boldsymbol{m}_{-i}) := [\mathcal{M}^U(m_i; \boldsymbol{m}_{-i})]_i - c_i m_i. \tag{2}$$

*If $u_i^r(m_i, \boldsymbol{m}_{-i})$, is quasi-concave for $m_i \geq m_i^u := \inf\{m_i | [\mathcal{M}^U(m_i; \boldsymbol{m}_{-i})]_i > 0\}$ and continuous in $\boldsymbol{m}_{-i}$, then a pure Nash equilibrium with $\boldsymbol{m}^{\boldsymbol{eq}}$ data contributions exists such that,*

$$u_i^r(\boldsymbol{m}^{\boldsymbol{eq}}) = [\mathcal{M}^U(\boldsymbol{m}^{\boldsymbol{eq}})]_i - c_i \boldsymbol{m}_i^{\boldsymbol{eq}} \geq [\mathcal{M}^U(m_i; \boldsymbol{m}_{-i}^{\boldsymbol{eq}})]_i - c_i m_i \; \forall m_i \geq 0. \tag{3}$$

The proof of Theorem 1 is found in Appendix C. Our new proof amends and simplifies that of Karimireddy et al. (2022), as we show $u_i^r$ must only be quasi-concave in $m_i$.

**Remark 1.** *Under only mild assumptions on the utility provided by mechanism $\mathcal{M}$ (feasibility, quasi-concavity, & continuity w.r.t data $m$), participating devices reach an equilibrium on local data usage for federated training $\boldsymbol{m}^{\boldsymbol{eq}}$. Deviating from such equilibrium contribution $\boldsymbol{m}_i^{\boldsymbol{eq}}$ results in a decrease in utility for device $i$ (Equation 3).*

In order for a mechanism $\mathcal{M}$ to provide data and output utility, via model-accuracy rewards (Equation 1), we must define a relationship between accuracy and data.

**Accuracy-Data Relationship**. Data reigns supreme when it comes to model performance; model accuracy improves as the quantity of training data increases, assuming consistency of data quality (Junqué de Fortuny et al., 2013; Tramer & Boneh, 2020; Zhu et al., 2012). Empirically, one finds that accuracy of a model is both concave and non-decreasing with respect to the amount of data used to train it (Sun et al., 2017). Training of an ML model generally adheres to the law of diminishing returns: improving model performance by training on more data is increasingly fruitless once the amount of training data is already large.

**Assumption 1.** *Accuracy function $\hat{a}_i(m)$ is continuous, non-decreasing, and concave w.r.t data $m$. Bounded accuracy $a_i(m) := \max\{\hat{a}_i(m), 0\}$ has a root at 0.*

For a given learning task, each device $i$ has a unique optimal attainable accuracy $a_{opt}^i := a_{opt}(\mathcal{D}_i) \in [0, 1)$ given its data distribution $\mathcal{D}_i$. Assumption 1 captures the empirical relationship between ML model accuracy $\hat{a}_i(m) : \mathbb{Z}^+ \to (-\infty, a_{opt}^i)$ and data $m$ in the wild: (**continuous & non-decreasing**) accuracy never decreases with more data and (**concavity**) accuracy experiences diminishing returns with more data. Since negative accuracy is impossible, we define $a_i(m) := \max\{\hat{a}_i(m), 0\}$. Contrary to previous work, Karimireddy et al. (2022), accuracy $a_i(m)$ is different for each device.

**Remark 2** (Attainability). *An accuracy function $\hat{a}_i(m)$ which satisfies Assumption 1 is:*

$$\hat{a}_i(m) := a_{opt}^i - \frac{\sqrt{2k(2 + \log(m/k))} + 4}{\sqrt{m}}. \tag{4}$$

*Our theory allows general $\hat{a}_i(m)$ as long as Assumption 1 is satisfied. However, like Karimireddy et al. (2022), we use $\hat{a}_i(m)$ as defined in Equation 4 for experiments. $k > 0$ denotes the hypothesis class complexity. Equation 4 is rooted in a generalization bound found in Appendix C.1.*

**Remark 3** (Server Accuracy). *The central server $C$ has its own accuracy function $\hat{a}_C(m)$ with optimal attainable accuracy $\bar{a}_{opt} := \mathbb{E}_{i \in [n]}\left[a_{opt}^i\right] = \sum_{i=1}^n \frac{m_i}{\sum_j m_j} a_{opt}^i$,*

$$\hat{a}_C(m) := \bar{a}_{opt} - \frac{\sqrt{2k(2 + \log(m/k))} + 4}{\sqrt{m}}. \tag{5}$$

**Remark 4** (Heterogeneous Distributions). *Instead of assuming that each device's local data is independently chosen from a common distribution (known as the IID setting), we generalize to the heterogeneous and non-IID setting. This is a major novelty of our work, as previous mechanisms, like Karimireddy et al. (2022), focus on identical device data distributions.*

## 4 MODELING REALISTIC UTILITY

Utility powers the performance of a mechanism. If utility is modeled incorrectly, *e.g.,* unrealistically, a mechanism will guide participants towards unrealistic and suboptimal results.

**A More Generalized Accuracy Payoff**. Unlike previous federated mechanisms, such as Karimireddy et al. (2022); Zhan et al. (2020b; 2021), we introduce a non-linear accuracy payoff compositional function $\phi_i(a(m)) : [0, 1) \to \mathbb{R}_{\geq 0}$, which allows for a flexible definition of the utility device $i$ receives from having a model with accuracy $a(m)$. In Karimireddy et al. (2022), it is assumed that the outer function $\phi_i$ is linear, $\phi_i(a(m)) = a(m)$, for all devices, which is restrictive. For example, accuracy improvement from $48\%$ to $49\%$ should be rewarded much differently than $98\%$ to $99\%$. Therefore, we generalize the outer function $\phi_i$ to be a convex and increasing function (which includes the linear case). These requirements (summarized in Assumption 2) ensure that increasing accuracy leads to enhanced utility for rational devices.

**Assumption 2.** $\phi_i(a_i(m)) : [0, a_{opt}^i) \to \mathbb{R}_{\geq 0}$ *is continuous and non-decreasing for each device $i$. The outer function $\phi_i(a)$ is convex and strictly increasing w.r.t $a$ ($\phi_i(0) = 0$). The compositional function $\phi_i(a_i(m))$ remains concave and strictly increasing w.r.t $m$ ($\forall m$ such that $\hat{a}_i(m) \geq 0$).*

Many **realistic choices** for $\phi_i$ which satisfy Assumption 2 exist. For $a_i(m)$ as in Equation 4, one reasonable choice is $\phi_i(a) = \frac{1}{(1-a)^2} - 1$. This choice captures how utility increasingly grows as accuracy approaches $100\%$, especially compared to the linear relationship $\phi_i(a) = a$. After defining the relationship between accuracy and utility, we can now formally define server and device utility.

**Defining Server Utility**. *The overarching goal for a central server $C$ is to attain a high-performing ML model from federated training.* As discussed at the beginning of Section 3, ML model accuracy generally improves as the total amount of data contributions $\boldsymbol{m}$ increase. Therefore, server utility $u_C$ is a function of model accuracy $a_C$ (Equation 5) which in turn is a function of data contributions,

$$u_C(\boldsymbol{m}) := p_m \cdot \phi_C\left(a_C\left(\sum \boldsymbol{m}\right)\right). \quad (6)$$

The fixed parameter $p_m \in (0, 1]$ denotes the central server's profit margin (percentage of utility kept by the central server). Since $p_m$ is fixed, server utility in Equation 6 is maximized when $a_C \to 100\%$. However, server accuracy is upper bounded by the optimal attainable accuracy $\bar{a}_{opt} \in [0, 1)$. Thus, to maximize Equation 6, the server wants to push $a_C \to \bar{a}_{opt} \approx 100\%$. Accomplishing this requires closing the

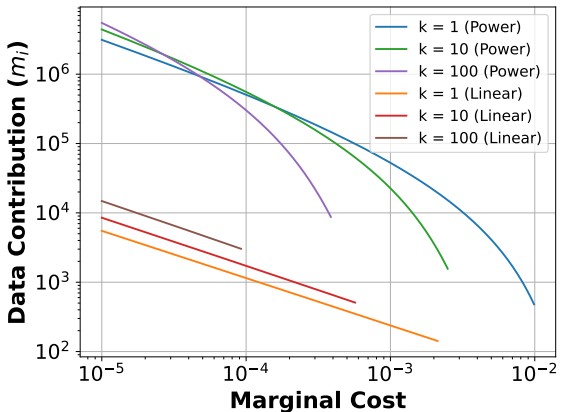

Figure 2: **Local Optimal Data Contribution for Varying Payoff Functions.** We compare optimal data contribution across different payoff functions. Realistic power payoff functions, $\phi_i(a) = \frac{1}{(1-a)^2} - 1$, result in greater optimal contribution compared to linear payoff functions, $\phi_i(a) = a$. We define $\hat{a}_i(m)$ as in Equation 4, with $a_{opt}^i = 0.95$ and multiple $k$ values.

gap between both (**i**) $a_C$ and $\bar{a}_{opt}$ as well as (**ii**) $\bar{a}_{opt}$ and $100\%$. The gap in (**i**) can be reduced by increasing the total amount of contributions $\sum \boldsymbol{m} \to \infty$ (via Equation 5). The gap in (**ii**) can shrink by receiving more contributions from devices $i$ with high optimal attainable accuracies $a_{opt}^i$ (Remark 3). Thus, an optimal mechanism $\mathcal{M}$ from the server's viewpoint *would be one which incentivizes more data used for federated contributions, with a larger proportion coming from devices $i$ with greater $a_{opt}^i$.* Finally, we note that only $p_m$ of the total collected utility is kept by the server in Equation 6. The value of $p_m$ is posted by the central server, and thus known by all devices, during the participation phase (Figure 1). In Section 5, we detail how our mechanism distributes the remaining $(1 - p_m)$ utility as a reward in proportion to how much data each participating device contributes.

**Defining Local Device Utility**. The utility $u_i$ for each device $i$ is a function of data contribution: devices determine how many data points $m$ to collect in order to balance the benefit of model accuracy

$\phi_i(a(m))$ versus the costs of data collection $-c_i m$. Thus, device utility is mathematically defined as,

$$u_i(m) = \phi_i(a_i(m)) - c_i m. \tag{7}$$

**Theorem 2** (Optimal Local Data Collection). *Consider device $i$ with marginal cost $c_i$, accuracy function $\hat{a}_i(m)$ satisfying Assumption 1, and accuracy payoff $\phi_i$ conforming to Assumption 2. Then the optimal amount of data $m_i^o$ device $i$ should collect is*

$$m_i^o = \begin{cases} 0 & \textit{if } \max_{m_i \geq 0} u_i(m_i) \leq 0 \quad \textit{else,} \\ m^*, \ \textit{s.t. } \phi_i'(\hat{a}_i(m^*)) \cdot \hat{a}_i'(m^*) = c_i. \end{cases} \tag{8}$$

Theorem 2 details how much data a device collects when *training on its own and thereby not participating in a federated training scheme*. We plot utility curves for both non-linear and linear accuracy payoff functions, with varying marginal costs $c_i$, in Figure 5. Figure 5c shows how utility peaks at a negative value when $c_i$ becomes too large. In this case, as shown in Theorem 2, devices do not contribute data. We defer proof of Theorem 2 to Appendix C.

## 5 REALFM: A STEP TOWARDS REALISTIC FEDERATED MECHANISMS

We reiterate that the goal of our proposed REALFM mechanism $\mathcal{M}_R$ (Algorithm 1) is to design a reward protocol, with model-accuracy $a^r$ and monetary $R$ rewards, such that rational devices choose to participate and contribute more data than is locally optimal (Theorem 2) in exchange for improved device utility. Denote the *utility* that REALFM provides to each device $i$ as $\mathcal{M}_{R,i}^U := [\mathcal{M}_R^U(\boldsymbol{m})]_i$.

$$\mathcal{M}_R(m_i; \boldsymbol{m}_{-i}) := \begin{cases} \big(a_i(m_i),\ 0\big) & \text{if } m_i \leq m_i^o, \\ \big(a_i(m_i^o) + \gamma_i(m_i),\ r(\boldsymbol{m})(m_i - m_i^o)\big) & \text{if } m_i \in [m_i^o, m_i^*], \\ \big(a_C(\sum \boldsymbol{m}),\ r(\boldsymbol{m})(m_i - m_i^o)\big) & \text{if } m_i \geq m_i^*. \end{cases} \tag{9}$$

$$\mathcal{M}_{R,i}^U := \begin{cases} \phi_i\big(a_i(m_i)\big) & \text{if } m_i \leq m_i^o, \\ \phi_i\big(a_i(m_i^o) + \gamma_i(m_i)\big) + r(\boldsymbol{m})(m_i - m_i^o) & \text{if } m_i \in [m_i^o, m_i^*], \\ \phi_i\big(a_C(\sum \boldsymbol{m})\big) + r(\boldsymbol{m})(m_i - m_i^o) & \text{if } m_i \geq m_i^*. \end{cases} \tag{10}$$

As described mathematically above, RE-ALFM eliminates free-riding by returning a model with accuracy equivalent to local training $a_i(m_i)$ for a device $i$ which does not contribute more than what is locally optimal ($m_i \leq m_i^o$). This can be accomplished in practice by carefully degrading the final model with noisy perturbations (or continued training on a skewed subset of data). Importantly, this process ensures Individually Rationality (IR), as devices receive at least as good performance as if they had trained by themselves (**incentivizing participation**). Devices which contribute more than what is locally optimal receive boosted accuracy $\gamma_i(m_i)$ and monetary rewards $r(\boldsymbol{m})(m_i - m_i^o)$ up to a new federated equilibrium $m_i^*$ (detailed in Theorem 3). This process ensures that devices are incentivized to contribute more than what is locally optimal (**incentivizing contributions**). Finally, if devices contribute more than the new equilibrium, the server can only provide the model accuracy equivalent to the fully trained global model $a_C(\sum \boldsymbol{m})$ as well as monetary rewards $r(\boldsymbol{m})(m_i - m_i^o)$

---

**Algorithm 1** REALFM

**Input:** Data contributions $\boldsymbol{m}$, marginal costs $\boldsymbol{c}$, profit margin $p_m$, payoff/shaping/accuracy functions $\phi/\gamma/a$, $h$ local steps, $T$ total iterations, total epochs $E$, initial parameters $\boldsymbol{w}^1$, loss $\ell$, and step-size $\eta$.
**Output:** Model accuracy $a_i^r$ and reward $R_i$.
$s_i \leftarrow m_i / \sum_{j=1}^n m_j$
**for** $t = 1, \ldots, T$ **do**
    Server distributes $\boldsymbol{w}^t$ to all devices
    **for** $h$ local steps, each device $i$ **in parallel do**
        $\boldsymbol{w}_i^{t+1} \leftarrow \text{ClientUpdate}(i, \boldsymbol{w}^t)$
    $\boldsymbol{w}^{t+1} \leftarrow \sum_j s_j \boldsymbol{w}_i^{t+1}$
$r(\boldsymbol{m}) \leftarrow (1 - p_m) \frac{\phi_C\big(a_C(\sum_j m_j)\big)}{\sum_j m_j}$
**for** $i = 1$ to $n$ **do**
    Compute $m_i^o$ and $m_i^*$ using $a_i, c_i, \phi_i$, and $\gamma_i(m_i)$
    Return $(a_i^r, R_i)$ to device $i$ using Eq. 9
**ClientUpdate**$(i, \boldsymbol{w})$: $\mathcal{B} \leftarrow$ batch $m_i$ data points
**for** each epoch $e = 1, \ldots, E$ **do**
    **for** batch $b \in \mathcal{B}$ **do**
        $\boldsymbol{w} \leftarrow \boldsymbol{w} - \eta \nabla \ell(\boldsymbol{w}; b)$

---

(**feasibility**). Below, we detail how the accuracy $\gamma_i$ and monetary $r(\boldsymbol{m})$ rewards are constructed to ensure devices receive greater utility even when they contribute more than what is locally optimal.

**Accuracy Rewards: Accuracy Shaping**. Accuracy shaping is responsible for incentivizing devices to collect more data than what is locally optimal $m_i^o$. The idea behind accuracy shaping is to incentivize device $i$ to use more data $m_i^* \geq m_i^o$ for federated training by providing a boosted model accuracy whose utility outstrips the marginal cost of collecting more data $c_i \cdot (m_i^* - m_i^o)$. Unlike Karimireddy et al. (2022), REALFM performs accuracy with non-linear accuracy payoffs $\phi$ ($\phi$ is assumed to be linear in Karimireddy et al. (2022)). To overcome the issues with non-linear $\phi$'s, we carefully construct an accuracy-shaping function $\gamma_i(m)$ for each device $i$. Devices which participate in $\mathcal{M}_R$ must share $c_i, \phi_i$ with the server.

> **Assumption 3.** *For a given set of device contributions $\boldsymbol{m}$, the maximum accuracy attained by the server must be greater than that of any single device, $a_C(\sum \boldsymbol{m}) \geq a_i(m_i^o) \; \forall i \in [n]$.*

Assumption 3 states that the globally-trained model outperforms any locally-trained model from the participating devices. This is valid if the direction of descent taken by the server is beneficial for all participating devices (*i.e.,* the inner product between local gradients and the aggregated gradient is positive). This assumption may be violated in certain non-iid settings, where a participating device's local data distribution greatly differs from the majority of devices. We note that our experiments use various non-iid data distributions, none of which violate Assumption 3.

> **Theorem 3** (Accuracy Shaping Guarantees). *Consider a device $i$ with marginal cost $c_i$ and accuracy payoff function $\phi_i$ satisfying Assumptions 2 and 3. Denote device $i$'s optimal local data contribution as $m_i^o$ and its subsequent accuracy $\bar{a}_i := a_i(m_i^o)$. Define the derivative of $\phi_i(a)$ with respect to $a$ as $\phi_i'(a)$. For any $\epsilon \to 0^+$ and marginal server reward $r(\boldsymbol{m}) \geq 0$, device $i$ has the following accuracy-shaping function $\gamma_i(m)$ for $m \geq m_i^o$,*
>
> $$\gamma_i := \begin{cases} \frac{-\phi_i'(\bar{a}) + \sqrt{\phi_i'(\bar{a})^2 + 2\phi_i''(\bar{a}_i)(c_i - r(\boldsymbol{m}) + \epsilon)(m - m_i^o)}}{\phi_i''(\bar{a}_i)}, \\ \frac{(c_i - r(\boldsymbol{m}) + \epsilon)(m - m_i^o)}{\phi_i'(\bar{a}_i)} \quad \text{if } \phi_i''(\bar{a}_i) = 0 \end{cases} \tag{11}$$
>
> *Given the defined $\gamma_i(m)$, the following inequality is satisfied for $m \in [m_i^o, m_i^*]$,*
>
> $$\phi_i(\bar{a}_i + \gamma_i(m)) - \phi_i(\bar{a}_i) > (c_i - r(\boldsymbol{m}))(m - m_i^o). \tag{12}$$
>
> *Now, $m_i^* := \{m \geq m_i^o \mid a_C(m + \sum_{j \neq i} m_j) = \bar{a}_i + \gamma_i(m)\}$ is the optimal contribution for each device $i$. Device $i$'s data contribution increases $m_i^* \geq m_i^o$ for any contribution $\boldsymbol{m}_{-i}$.*

Theorem 3 (proof in Appendix C) defines an accuracy-shaping function $\gamma_i$ (Equation 11) that ensures devices receive more gain in utility than loss by contributing more than is locally optimal (Equation 12) up to a feasible equilibrium $m_i^*$ (the server cannot provide an accuracy beyond $a_C(\sum \boldsymbol{m})$).

> **Remark 5.** *For a linear accuracy payoff, $\phi_i(a) = wa$ for $w > 0$, Equation 11 relays $\gamma_i = \frac{(c_i - r(\boldsymbol{m}) + \epsilon)(m - m_i^o)}{w}$. We recover the accuracy-shaping function, Equation (13), in Karimireddy et al. (2022) with their no-reward $r(\boldsymbol{m}) = 0$ and $w = 1$ linear setting. Thus, our accuracy-shaping function generalizes the one in Karimireddy et al. (2022).*

> **Remark 6** (Proportional Shaping). *When local accuracy $\bar{a}_i$ is low, the values of $\phi_i'(\bar{a}_i), \phi_i''(\bar{a})$ are smaller (Assumption 2) and thus $\gamma_i(m)$ grows faster w.r.t $m$ (Equation 11). Thus, for a low-accuracy device $i$, $\gamma_i$ will reach its upper accuracy limit at a smaller optimal contribution $m_i^*$. The result is proportional shaping: contributions are incentivized in proportion to device performance.*

**Monetary Rewards**. As detailed in Section 4, the server keeps a fraction $p_m$ of its utility gained at the end of training (Equation 6). REALFM disperses the remaining $1 - p_m$ utility as a marginal monetary reward $r(\boldsymbol{m})$ for each data point contributed more than locally optimal,

$$r(\boldsymbol{m}) := (1 - p_m) \cdot \phi_C \left( a_C \left( \sum \boldsymbol{m} \right) \right) / \sum \boldsymbol{m} \longrightarrow R_i := r(\boldsymbol{m})(m_i - m_i^o). \tag{13}$$

The marginal monetary reward $r(\boldsymbol{m})$ is dynamic and depends upon the total amount of data used by devices during federated training. Therefore, $r(\boldsymbol{m})$ is unknown to devices when $\mathcal{M}_R$ is issued. However, the server computes and provides the monetary rewards once training is complete.

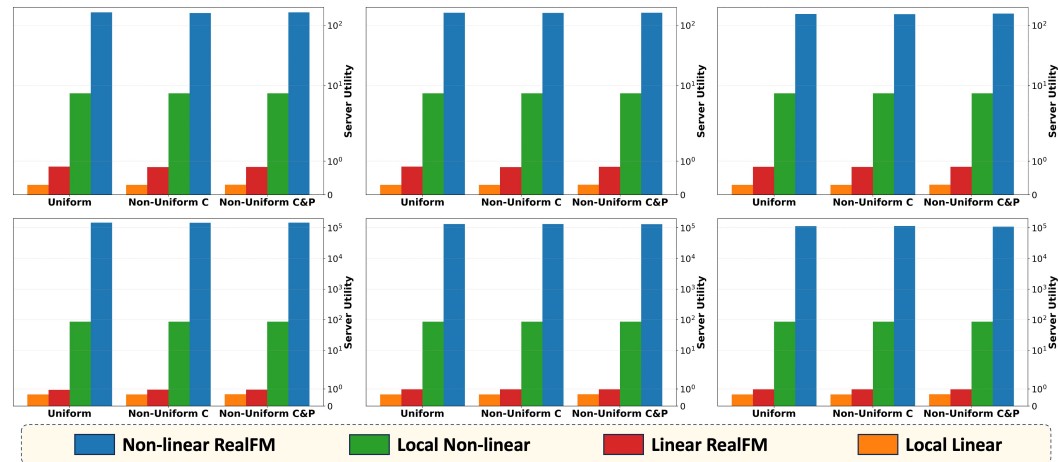

Figure 3: **Improved Server Utility on CIFAR-10 & MNIST.** REALFM increases server utility on CIFAR-10 (top row) and MNIST (bottom row) for 16 devices compared to baselines. REALFM achieves upwards of 5 magnitudes more utility than a FL version of Karimireddy et al. (2022), denoted as LINEAR REALFM, across both uniform and various heterogeneous Dirichlet data distributions (left: uniform, center: D-0.6, right: D-0.3) as well as non-uniform costs (C) and accuracy payoff functions (P).

**Theorem 4** (Existence of Improved Equilibrium). *REALFM $\mathcal{M}_R$ (Equation 9) performs accuracy-shaping with $\gamma_i$ defined in Theorem 3 for each device $i \in [n]$ and some $\epsilon \to 0^+$. As such, $\mathcal{M}_R$ is Individually Rational (IR) and has a unique Nash equilibrium at which device $i$ will contribute $m_i^* \geq m_i^o$ updates, thereby eliminating the free-rider phenomena. Furthermore, since $\mathcal{M}_R$ is IR, devices are incentivized to participate as they gain equal to or more utility than by not participating.*

Since REALFM (**i**) returns model accuracy equivalent to local training if devices do not contribute more than what is locally optimal, and (**ii**) ensures that devices are provided improved utility when contributing more than locally optimal (Theorem 3), REALFM is IR. Furthermore, since $\mathcal{M}_R$ is feasible and the utility provided is continuous and quasi-concave (Equation 10), there exists a Nash equilibrium (Theorem 1). The use of accuracy shaping (Theorem 3) also ensures that devices contribute more data than is locally optimal, eliminating the free-rider effect. Our experiments provide empirical backing that REALFM indeed incentives devices to contribute more data and improves both device and server utility.

## 6 EXPERIMENTAL RESULTS

To test the efficacy of REALFM, we analyze how well it performs at (**i**) improving utility for the central server and devices, and (**ii**) increasing the amount of data contributions to federated training on image classification experiments. We perform experiments on CIFAR-10 (Krizhevsky et al., 2009) and MNIST (Deng, 2012).

**Experimental Baselines.** Few FL mechanisms eliminate the free-rider effect, with none doing so without sharing data. Therefore, we adapt the mechanism proposed by Karimireddy et al. (2022) as the baseline to compare against (we denote it as LINEAR REALFM). We also compare REALFM to a local training baseline where we measure the average device utility attained by devices if they did not participate in the mechanism. Server utility is inferred in this instance by using the average accuracy of locally trained models.

**Testing Scenarios.** We test REALFM and its baselines under homogeneous and heterogeneous device data distributions. In the heterogeneous case, we use two different Dirichlet distributions (parameters 0.6 and 0.3) to determine label proportions for each device Gao et al. (2022). These settings are

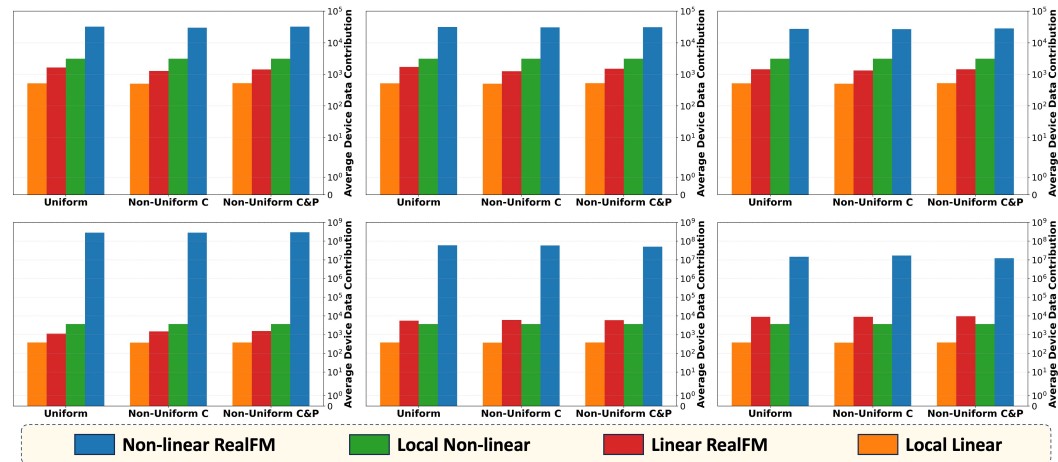

Figure 4: **Increased Federated Contribution on CIFAR-10 & MNIST.** REALFM incentivizes devices to use more local data during federated training on CIFAR-10 (top row) and MNIST (bottom row) for 16 devices compared to relevant baselines. REALFM achieves upwards of 4 magnitudes more federated contributions than LINEAR REALFM across both uniform and various heterogeneous Dirichlet data distributions (left: uniform, center: D-0.6, right: D-0.3) as well as non-uniform costs (C) and accuracy payoff functions (P).

denoted as $D - 0.6$ and $D - 0.3$ respectively. We also test REALFM under non-uniform marginal costs (C) and accuracy payoffs functions (P). Due to space constraints, additional experiments and details are in Appendix B.

**Increased Contributions to Federated Training.** Figure 4 showcases the power of REALFM, via its accuracy-shaping function, to incentivize devices to contribute more data points. Through its construction, REALFM's accuracy-shaping function incentivizes devices to use more data during federated training than locally optimal in exchange for greater utility. This is important for two reasons. First, incentivizing devices to contribute more than local training proves that the free-rider effect is not taking place. Second, higher data contribution lead to better-performing models and higher accuracies. This improves the utility for all participants. Overall, REALFM is superior at incentivizing contributions compared to state-of-the-art FL mechanisms.

**Improved Server and Device Utility.** REALFM leverages an improved reward mechanism (Equation 9) to boost the amount of contributions to federated training. The influx of data subsequently increases model performance (detailed in Section 3). This is backed up empirically: Figure 3 showcases upwards of 5 *magnitudes greater server utility* compared to state-of-the-art FL mechanisms. Devices participating in REALFM also boost utility *by over* 5 *magnitudes* (Figure 7). The improvement stems from (i) effective accuracy-shaping by REALFM and (ii) the use of non-linear accuracy payoff functions $\phi$, which more precisely map the benefit derived from an increase in model accuracy.

**Performance Under Non-Uniformities.** We find that non-uniform costs and accuracy payoff functions do not affect REALFM performance. However, as expected, REALFM performance slightly degrades as device datasets become more heterogeneous (Figures 3 & 4). In this setting, both local and federated accuracies are lower due to the difficulties of out-of-distribution generalization and model drift. Nevertheless, even under heterogeneous distributions, REALFM greatly outperforms all other baselines under both non-uniform costs and accuracy payoff functions as well as non-uniform (heterogeneous) device data distributions.

## 7 CONCLUSION AND IMPACT

Without proper incentives, modern FL frameworks fail to attract devices to participate. Even if devices do participate, current FL frameworks fall victim to the free-rider dilemma. REALFM is the first FL mechanism to simultaneously incentivize device participation and contribution in a realistic

manner. Unlike other FL mechanisms, REALFM utilizes a non-linear relationship between model accuracy and utility, allows heterogeneous data distributions, removes data sharing requirements, and models central server utility. Empirically, we show that REALFM's realistic utility and effective incentive structure, using a novel accuracy-shaping function, results in (i) improved server and device utility, (ii) increased federated contributions, and (iii) higher-performing models during federated training compared to its peer FL mechanisms.

**Impact Statement**. Edge devices have long been taken for granted within Federated Learning, often assumed to be at the beck and call of the central server. Devices are expected to provide the server with gradient updates computed on their own valuable local data, incurring potentially large computational and communication costs. All of this occurs without discussion between the server and devices over proper compensation for each device's data usage and work.

Our paper aims to produce realistic federated frameworks that benefit the server and participating devices. One overarching goal of our paper is to ensure that devices are properly incentivized (compensated) by the server for their participation in federated training. As detailed within our paper, incentivizing device participation and contribution also helps the server; model accuracy improves with greater quantity and diversity of gradient updates. Thus, the impact of our work lies in showing that incorporating equity within Federated Learning can indeed lead to a more desirable result for all parties involved. By adequately incentivizing and compensating edge devices for their time, work, and data, utility increases for both devices *and* the server.

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

# REALFM Appendix

## CONTENTS

## A  Notation & Related Work

Table 1: Notation Table for REALFM.

| Definition | Notation |
|:---:|:---:|
| Number of Devices | $n$ |
| Local FedAvg Training Steps | $h$ |
| Optimal Attainable Accuracy on Learning Task | $a_{opt}$ |
| Number of Data Points | $m$ |
| Total Data Point Contributions | $\boldsymbol{m}$ |
| Accuracy Function | $a(m)$ |
| Mechanism | $\mathcal{M}$ |
| Server Profit Margin | $p_m$ |
| Server Payoff Function | $\phi_C$ |
| Model Parameters | $\boldsymbol{w}$ |
| Marginal Cost for Device $i$ | $c_i$ |
| Payoff Function for Device $i$ | $\phi_i$ |
| Device $i$ Utility | $u_i$ |
| Data Distribution for Device $i$ | $\mathcal{D}_i$ |
| Local Optimal Device $i$ Utility | $u_i^0$ |
| Rewarded Mechanism Utility for Device $i$ | $u_i^r$ |
| Local Optimal Data Contribution for Device $i$ | $m_i^o$ |
| Mechanism Optimal Data Contribution for Device $i$ | $m_i^*$ |
| Mechanism Model Accuracy Reward | $a^r$ |
| Mechanism Monetary Reward | $R$ |
| Marginal Monetary Reward per Contributed Data Point | $r(\boldsymbol{m})$ |
| Accuracy-Shaping Function for Device $i$ | $\gamma_i$ |

**Federated Mechanisms (Continued).** As detailed in Section 2, there is a wide swath of mechanisms proposed for FL. The works (Zhan et al., 2021; Tu et al., 2022; Zeng et al., 2021; Ali et al., 2023) survey the different methods of incentives present in FL literature. The goal of the presented methods are solely to increase device participation within FL frameworks. The issues of free riding and increased data or gradient contribution are ignored. (Sim et al., 2020; Xu et al., 2021) design model rewards to meet fairness or accuracy objectives. In these works, as detailed below, devices receive models proportionate to the amount of data they contribute but are not incentivized to contribute more data. Our work seeks to incentivize devices to contribute more during training.

**Collaborative Fairness and Federated Shapley Value.** Collaborative fairness in FL (Lyu et al., 2020a;b; Xu & Lyu, 2020; Sim et al., 2020) is closely related to our paper. The works Lyu et al. (2020a;b); Xu & Lyu (2020); Sim et al. (2020) seek to fairly allocate models with varying performance depending upon how much devices contribute to training in FL settings. This is accomplished by determining a "reputation" (a measure of device contributions) for each device, using a hyperbolic sine function, to enforce devices converge to different models relative to their amount of contributions during FL training. Thus, devices who contribute more receive a higher-performing model than those who do not contribute much. There are a few key differences between this line of work and our own, namely: **(1)** Our mechanism incentivizes devices to both participate in training and increase their amount of contributions. There are no such incentives in collaborative fairness. **(2)** We model device and server utility. Unlike collaborative fairness, we do not assume that devices will always participate in FL training. **(3)** Our mechanism design *provably* eliminates the free-rider phenomena unlike collaborative fairness, since devices who try to free-ride receive the same model performance as they would on their own (Equations 10 and 9). **(4)** No monetary rewards are received by devices in current collaborative fairness methods.

Federated Shapley Value (FSV), first proposed in Wang et al. (2020), allows for estimation of the Shapley Value in a FL setting. This is crucial to appraise the data coming from each device and possibly pave the way for rewarding devices with important data during the training process. While our work allows devices to have heterogeneous data distributions, we do not perform data valuation to further fine-tune rewards for each device (*i.e.,* provide more accurate models or more

monetary rewards to devices who provide more valuable data). The overarching goal of our work is to incentivize and increase device participation, through mechanism design, within FL. However, further reward fine-tuning via FSV remains the subject of future research.

**Honest Devices.** Our setting assumes honest devices. They only store the rewarded model returned by the server after training. If devices are dishonest, slight alterations to the federated training scheme can be made. Namely, schemes that train varying-sized models such as (Arivazhagan et al., 2019; Diao et al., 2020; Hu et al., 2021; Li & Wang, 2019) can be implemented (where model sizes correspond to data contribution size) to alleviate honesty issues.

**Contribution Maximization**. The usage of a non-linear accuracy payoff function $\phi_i$ promotes increased contributions compared to a linear payoff (see Section 6). However, proof of contribution maximization for non-linear payoffs is not possible, as one cannot tightly bound the composition function $\phi(a + \gamma(m))$ as one can with the linear payoff $a + \gamma(m)$. Our accuracy-shaping function is still contribution maximizing when linear.

**Corollary 1.** *Mechanism $\mathcal{M}_R$ (Equation 9) is contribution-maximizing for linear accuracy payoffs. The proof follows Theorem 4.2 in Karimireddy et al. (2022).*

# B EXPERIMENTAL RESULTS CONTINUED

In this section we provide further details into how our image classification experiments were run as well as provide additional experiments. As a note, we ran each experiment three times, varying the random seeds. All bar plots in our paper showcase the mean results of the three experiments. In Figures 6 and 8, we plot test accuracies and error bars for CIFAR-10 & MNIST. The error bars are thin, as the results did not vary much between each of the three experiments. Finally, for simplicity and conservative results within all of our experiments, **we set the server's profit margin as $p_m = 1$ (greedy server)**. All experiments were run using a cluster of 2-4 GPUs shared across 8/16 CPUs. We use GeForce GTX 1080 Ti GPUs (11GB of memory) and the CPUs used are Xeon 4216.

## B.1 ADDITIONAL EXPERIMENTAL DETAILS

**Experimental Setup.** Within our experiments, both 8 and 16 devices train a ResNet18 and a small convolutional neural network for CIFAR-10 and MNIST respectively. We use Stochastic Gradient Descent for CIFAR-10 and Adam for MNIST during training. As detailed in Appendix B, we carefully select and tune $a_C(m)$ to match the empirical training results on both datasets as closely as possible. Once tuned, we select a fixed marginal cost $c_e$ of 2.5e-4 (4e-5) for CIFAR-10 (MNIST) on all baselines. When performing uniform cost experiments, each device uses $c_e$ as its marginal cost (and thus each device has the same amount of data). For non-uniform cost experiments, $c_e$ is the mean of a Gaussian distribution from which the marginal costs are sampled. Our uniform accuracy payoff is $\phi_i(a) = \frac{1}{(1-a)^2} - 1$ for each device $i$. For non-uniform payoff experiments, we set $\phi_i(a) = z_i\left(\frac{1}{(1-a)^2} - 1\right)$ where $z_i$ is uniformly sampled within $[0.9, 1.1]$.

**Experimental Process.** The experimental process involved careful tuning of our theoretical accuracy function $\hat{a}_C(m)$ in order to match the empirical accuracy results we found. In fact, for CIFAR-10 we use $\hat{a}_C(m)$ defined in Equation 4 with carefully selected values for $k$ and $\bar{a}_{opt}$ to precisely reflect the empirical results found for our CIFAR-10 training. For MNIST, however, an accuracy function of $\hat{a}_C(m) = \bar{a}_{opt} - 2\sqrt{k/m}$ was more reflective of the ease of training on MNIST. Overall, to ensure precise empirical results, we followed the following process for each experiment (for both CIFAR-10 and MNIST as well as for 8 and 16 devices):

1. Determine $\bar{a}_{opt}$ and $k$ such that $\hat{a}_C(m)$ is tuned to most precisely reflect our empirical results.
2. Select a uniform marginal cost $c_e$ low enough for non-zero amount of data to be used by devices.
3. For non-uniform experiments, draw a marginal cost at random from a Gaussian with mean $c_e$ and/or draw a payoff function $\phi_i$ with a $z_i$ sampled within $[0.9, 1.1]$ (detailed in Section 6).

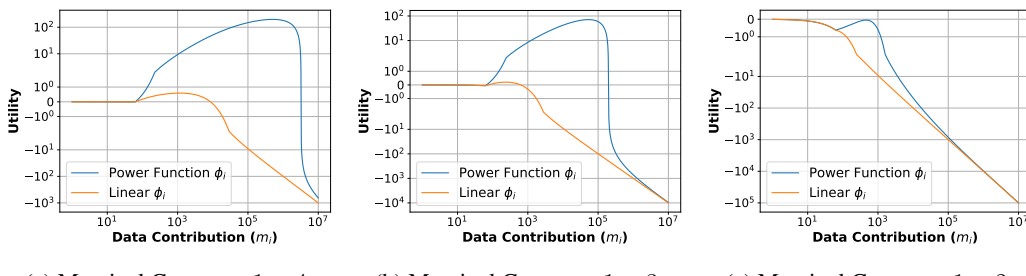

(a) Marginal Cost $c_i = 1\mathrm{e}{-4}$.  (b) Marginal Cost $c_i = 1\mathrm{e}{-3}$.  (c) Marginal Cost $c_i = 1\mathrm{e}{-2}$.

Figure 5: **Utility Functions for Varying Cost and Payoff Functions.** Using both linear, $\phi_i(a) = a$, and power, $\phi_i(a) = \frac{1}{(1-a)^2} - 1$, payoff functions, we compare how device utilities change with rising costs. Once marginal costs $c_i$ become too high, the utility is always negative and devices will not collect data for training. We use $\hat{a}_i(m)$ as defined in Equation 4, with $a_{opt}^i = 0.95$ and $k = 1$.

4. Given the marginal cost and $a_C(m)$, derive the locally optimal data $m_i$ for each device $i$.
5. Save an initial model for training.
6. Train this initial model locally on each device until convergence, generating $a_{local}$.
7. Using the initial model, train the model in a federated manner until convergence, generating $a_{fed}$.
8. Using the accuracy-shaping function, compute the amount of additional data contributions required to raise $a_{local}$ to $a_{fed}$. *This is the incentivized amount of added contributions.*

We also use the expected payoff function for the central server: $\phi_C = 1/(1-a)^2 - 1$. We note that a learning rate scheduler is used for CIFAR-10 but not MNIST. Below, we detail the hyper-parameters used in our CIFAR-10 and MNIST experiments.

Table 2: Hyper-parameters for CIFAR-10 Experiments.

| Model | Batch Size | Learning Rate | Marginal Cost $c_e$ | $a_{opt}$ | $k$ | Epochs | Local FedAvg Steps $h$ |
|---|---|---|---|---|---|---|---|
| ResNet18 | 128 | 0.05 | 2.5e-4 | 0.95 | 10 | 100 | 6 |

Table 3: Hyper-parameters for MNIST Experiments.

| Model | Batch Size | Learning Rate | Marginal Cost $c_e$ | $a_{opt}$ | $k$ | Epochs | Local FedAvg Steps $h$ |
|---|---|---|---|---|---|---|---|
| CNN | 128 | 1e-3 | 4e-5 | 0.9975 | 0.25 | 100 | 6 |

### B.2 ADDITIONAL EXPERIMENTAL RESULTS

It is interesting to note how well REALFM performs on MNIST (in terms of the vastly improved utility seen in Figures 3 and 9) while FedAvg only improves model accuracy by a mere couple of percentage points. The reason stems from the payoff function $\phi_i$ which heavily rewards models that have accuracies close to 100%. This scenario is rational in real-world settings. Competing companies in industry will all likely have models which are high-performing (above 95% accuracy). Since the competition is stiff, companies with the best model performance will likely attract the most customers since their product is the best. Therefore, the utility for achieving model performance close to 100% should become larger and larger as one gets closer to 100%.

#### B.2.1 ADDITIONAL 16 DEVICE EXPERIMENTS

In Figure 6 we provide the additional accuracy curves for our 16 device experiments as well as device utility comparisons for REALFM versus its baseline algorithms.

#### B.2.2 8 DEVICE EXPERIMENTS

Below we provide CIFAR-10 and MNIST results for 8 devices. These plots mirror those shown in Section 6 for 16 devices. In both cases, we find that utility sharply improves for the central server

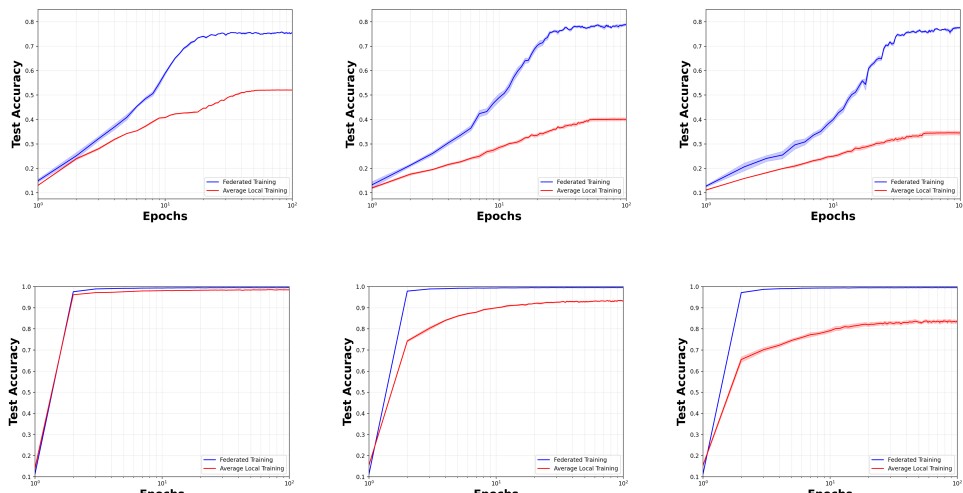

Figure 6: **Accuracy Comparison on CIFAR-10 & MNIST.** We plot the difference between local training (red line) and federated training (blue line) on CIFAR-10 (top row) and MNIST (bottom row) for 16 devices with uniform marginal costs and payoff functions. As expected, federated training always outperforms local training. Both uniform and various heterogeneous Dirichlet data distributions are shown above (left: uniform, center: D-0.6, right: D-0.3).

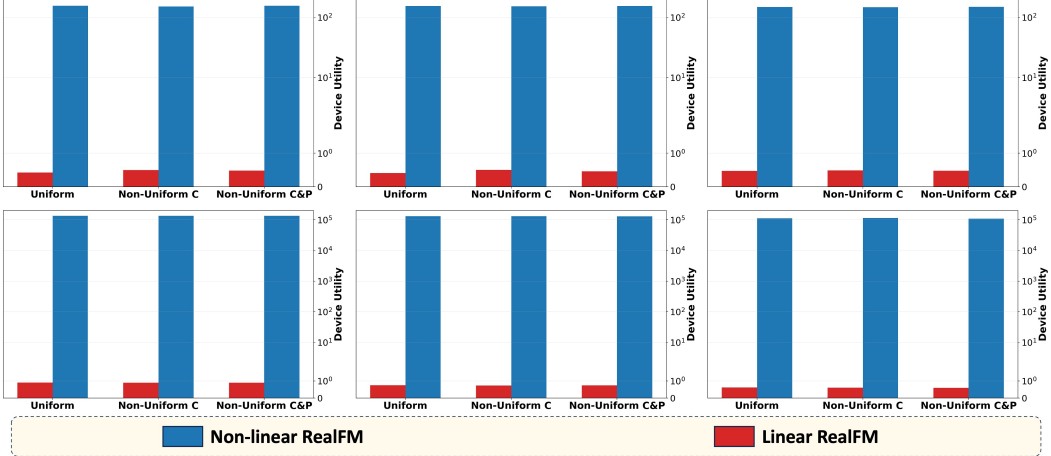

Figure 7: **Improved Device Utility on CIFAR-10 & MNIST.** REALFM increases device utility on CIFAR-10 (top row) and MNIST (bottom row) for 16 devices compared to baselines. REALFM achieves upwards of 5 magnitudes more utility than a FL version of Karimireddy et al. (2022), denoted as LINEAR REALFM, across both uniform and various heterogeneous Dirichlet data distributions (left: uniform, center: D-0.6, right: D-0.3) as well as non-uniform costs (C) and accuracy payoff functions (P).

and participating devices. Data contribution also improves for CIFAR-10 and MNIST. Under all scenarios REALFM performs the best compared to all other baselines. First, we start with the test accuracy curves for both datasets.

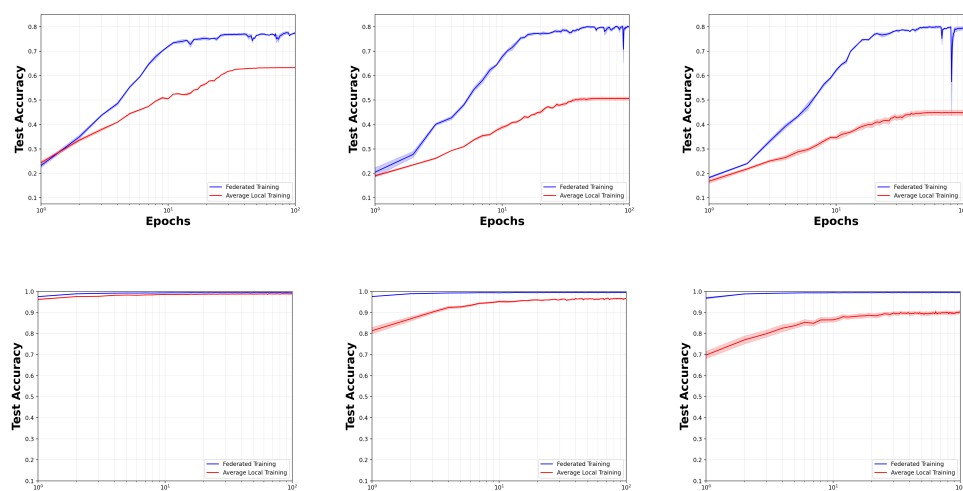

Figure 8: **Accuracy Comparison on CIFAR-10 & MNIST.** We plot the difference between local training (red line) and federated training (blue line) on CIFAR-10 (top row) and MNIST (bottom row) for 8 devices with uniform marginal costs and payoff functions. As expected, federated training always outperforms local training. Both uniform and various heterogeneous Dirichlet data distributions are shown above (left: uniform, center: D-0.6, right: D-0.3).

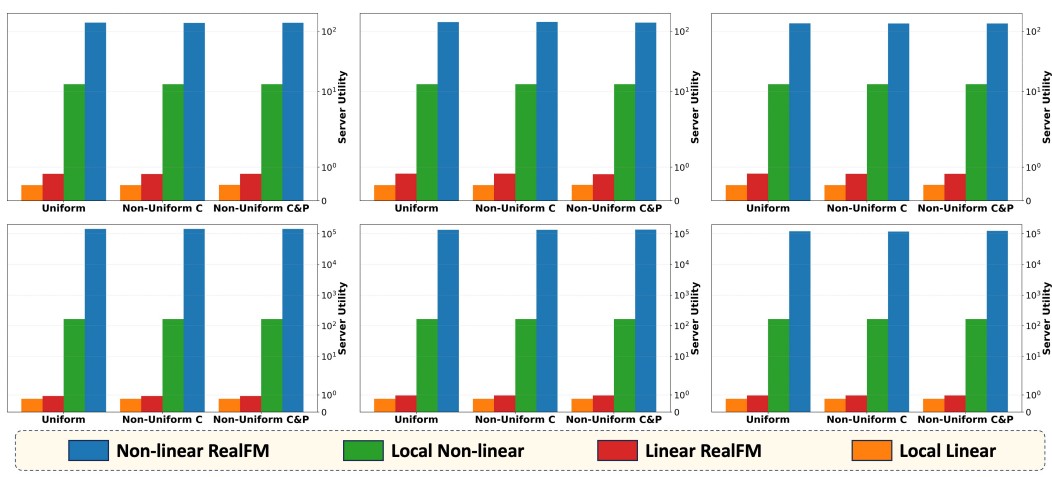

Figure 9: **Improved Server Utility on CIFAR-10 & MNIST.** REALFM increases server utility on CIFAR-10 (top row) and MNIST (bottom row) for 8 devices compared to baselines. REALFM achieves upwards of 5 magnitudes more utility than a FL version of Karimireddy et al. (2022), denoted as LINEAR REALFM, across both uniform and various heterogeneous Dirichlet data distributions (left: uniform, center: D-0.6, right: D-0.3) as well as non-uniform costs (C) and accuracy payoff functions (P).

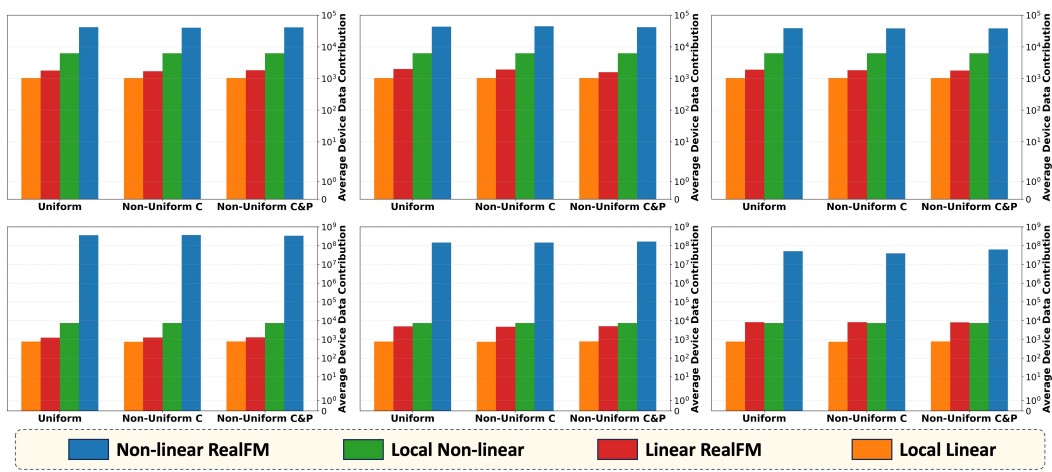

Figure 10: **Increased Federated Contribution on CIFAR-10 & MNIST.** REALFM incentivizes devices to use more local data during federated training on CIFAR-10 (top row) and MNIST (bottom row) for 8 devices compared to relevant baselines. REALFM achieves upwards of 4 magnitudes more federated contributions than a FL version of Karimireddy et al. (2022), denoted as LINEAR REALFM, across both uniform and various heterogeneous Dirichlet data distributions (left: uniform, center: D-0.6, right: D-0.3) as well as non-uniform costs (C) and accuracy payoff functions (P).

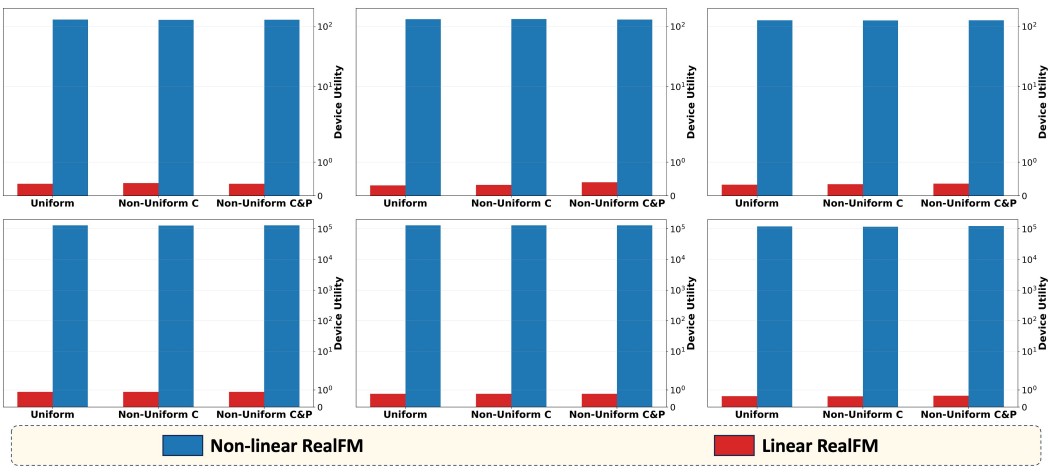

Figure 11: **Improved Device Utility on CIFAR-10 & MNIST.** REALFM increases device utility on CIFAR-10 (top row) and MNIST (bottom row) for 8 devices compared to baselines. REALFM achieves upwards of 5 magnitudes more utility than a FL version of Karimireddy et al. (2022), denoted as LINEAR REALFM, across both uniform and various heterogeneous Dirichlet data distributions (left: uniform, center: D-0.6, right: D-0.3) as well as non-uniform costs (C) and accuracy payoff functions (P).

## C    PROOF OF THEOREMS

**Theorem 1.** *Consider a feasible mechanism $\mathcal{M}$ returning utility $[\mathcal{M}^U(m_i; \boldsymbol{m}_{-i})]_i$ to device $i$ ($[\mathcal{M}^U(0; \boldsymbol{m}_{-i})]_i = 0$). Define the utility of a participating device $i$ as,*

$$u_i^r(m_i; \boldsymbol{m}_{-i}) := [\mathcal{M}^U(m_i; \boldsymbol{m}_{-i})]_i - c_i m_i. \tag{14}$$

*If $u_i^r(m_i, \boldsymbol{m}_{-i})$, is quasi-concave for $m_i \geq m_i^u := \inf\{m_i \mid [\mathcal{M}^U(m_i; \boldsymbol{m}_{-i})]_i > 0\}$ and continuous in $\boldsymbol{m}_{-i}$, then a pure Nash equilibrium with $\boldsymbol{m}^{eq}$ data contributions exists such that,*

$$u_i^r(\boldsymbol{m}^{eq}) = [\mathcal{M}^U(\boldsymbol{m}^{eq})]_i - c_i \boldsymbol{m}_i^{eq} \geq u_i^r(m_i; \boldsymbol{m}_{-i}^{eq}) = [\mathcal{M}^U(m_i; \boldsymbol{m}_{-i}^{eq})]_i - c_i m_i \text{ for } m_i \geq 0. \tag{15}$$

*Proof.* We start the proof by examining two scenarios.

**Case 1:** $\max_{m_i} u_i^r(m_i; \boldsymbol{m}_{-i}) \leq 0$.

In the case where the marginal cost of producing updates $-c_i m_i$ is so large that the device utility $u_i$ will always be non-positive, the best response $[B(\boldsymbol{m})]_i$ given a set of contributions $\boldsymbol{m}$ for device $i$ is,

$$[B(\boldsymbol{m})]_i = \arg\max_{m_i \geq 0} u_i^r(m_i; \boldsymbol{m}_{-i}) = 0. \tag{16}$$

As expected, device $i$ will not perform any updates $\boldsymbol{m}_i^{eq} = 0$. Therefore, Equation 3 is fulfilled since as $\max_{m_i} u_i^r(m_i; \boldsymbol{m}_{-i}) \leq 0$ we see,

$$[\mathcal{M}^U(\boldsymbol{m}^{eq})]_i - c_i \boldsymbol{m}_i^{eq} = [\mathcal{M}^U(0; \boldsymbol{m}_{-i})]_i - c_i(0) = 0 \geq [\mathcal{M}^U(m_i; \boldsymbol{m}_{-i}^{eq})]_i - c_i m_i \text{ for all } m_i \geq 0. \tag{17}$$

**Case 2:** $\max_{m_i} u_i(m_i; \boldsymbol{m}_{-i}) > 0$.

Denote $m_i^u := \inf\{m_i \mid [\mathcal{M}^U(m_i; \boldsymbol{m}_{-i})]_i > 0\}$. On the interval of integers $m_i \in [0, m_i^u]$, device $i$'s utility is non-positive,

$$u_i^r(m_i; \boldsymbol{m}_{-i}) = [\mathcal{M}^U(m_i; \boldsymbol{m}_{-i})]_i - c_i m_i = -c_i m_i \leq 0, \quad \forall m_i \in [0, m_i^u]. \tag{18}$$

For $m_i \geq m_i^u$, we have that $u_i(m_i; \boldsymbol{m})$ is quasi-concave. Let the best response for a given set of contributions $\boldsymbol{m}_{-i}$ for device $i$ be formally defined as,

$$[B(\boldsymbol{m})]_i := \arg\max_{m_i \geq 0} u_i^r(m_i; \boldsymbol{m}_{-i}) = \arg\max_{m_i \geq 0} [\mathcal{M}^U(m_i; \boldsymbol{m}_{-i})]_i - c_i m_i. \tag{19}$$

Suppose there exists a fixed point $\tilde{\boldsymbol{m}}$ to the best response, $\tilde{\boldsymbol{m}} \in B(\tilde{\boldsymbol{m}})$. This would mean that $\tilde{\boldsymbol{m}}$ is an equilibrium since by Equation 19 we have for any $m_i \geq 0$,

$$[\mathcal{M}^U(\tilde{m}_i; \tilde{\boldsymbol{m}}_{-i})]_i - c_i \tilde{m}_i \geq [\mathcal{M}^U(m_i; \boldsymbol{m}_{-i})]_i - c_i m_i. \tag{20}$$

Thus, now we must show that $B$ has a fixed point (which is subsequently an equilibrium). To do so, we first determine a convex and compact search space. As detailed in Case 1, $u_i^r(0, \boldsymbol{m}_{-i}) = 0$. Therefore, we can bound $0 \leq \max_{m_i} u_i^r(m_i, \boldsymbol{m}_{-i})$. Since $\mathcal{M}$ is feasible (Definition 1), $\mathcal{M}(\boldsymbol{m})$ is bounded above by $\mathcal{M}_{max}^U$. Thus, we find

$$0 \leq \max_{m_i} u_i^r(m_i; \boldsymbol{m}_{-i}) \leq \mathcal{M}_{max}^U - c_i m_i. \tag{21}$$

Rearranging yields $m_i \leq \mathcal{M}_{max}^U / c_i$. Since $m_i \geq m_i^u$, we can restrict our search space to $\mathcal{C} := \prod_j [m_j^u, \mathcal{M}_{max}^U / c_j] \subset \mathbb{R}^n$, where our best response mapping is now over $B : \mathcal{C} \to 2^\mathcal{C}$.

**Lemma 1** (Kakutani's Theorem). *Consider a multi-valued function $F : \mathcal{C} \to 2^\mathcal{C}$ over convex and compact domain $\mathcal{C}$ for which the output set $F(\boldsymbol{m})$ (i) is convex and closed for any fixed $\boldsymbol{m}$, and (ii) changes continuously as we change $\boldsymbol{m}$. For any such $F$, there exists a fixed point $\boldsymbol{m}$ such that $\boldsymbol{m} \in F(\boldsymbol{m})$.*

Since within this interval of $m_i$ $u_i^r(m_i, \boldsymbol{m}_{-i})$ is quasi-concave, $B(\boldsymbol{m})$ must be continuous in $\boldsymbol{m}$ (from Acemoglu & Ozdaglar (2009)). Now by applying Lemma 1, Kakutani's fixed point theorem, there exists such a fixed point $\tilde{\boldsymbol{m}}$ such that $\tilde{\boldsymbol{m}} \in B(\tilde{\boldsymbol{m}})$ where $\tilde{\boldsymbol{m}}_i \geq m_i^u$. Since $\max_{m_i} u_i^r(m_i; \boldsymbol{m}_{-i}) > 0$ and $u_i^r(m_i; \boldsymbol{m}_{-i}) \leq 0$ for $m_i \in [0, m_i^u]$, $\tilde{\boldsymbol{m}}_i$ is certain to not fall within $[0, m_i^u]$ $\forall i$ due to the nature of the arg max in Equation 19. Therefore, Equation 20 holds as the fixed point $\tilde{\boldsymbol{m}}$ exists and is the equilibrium of $\mathcal{M}$. $\qquad\square$

**Theorem 2.** *Consider a device $i$ with marginal cost per data point $c_i$, accuracy function $\hat{a}_i(m)$ satisfying Assumption 1, and accuracy payoff $\phi_i$ satisfying Assumption 2. This device will collect the following optimal amount of data $m_i^o$:*

$$m_i^o = \begin{cases} 0 & \text{if } \max_{m_i \geq 0} u_i(m_i) \leq 0, \\ m^*, \text{ such that } \phi_i'(\hat{a}_i(m^*)) \cdot \hat{a}_i'(m^*) = c_i & \text{else.} \end{cases} \tag{22}$$

*Proof.* Let $m_0 := \sup\{m \mid \hat{a}_i(m) = 0\}$ (the point where $a_i(m)$ begins to increase from 0 and become equivalent to $\hat{a}_i(m)$). Thus, $\forall m_i > m_0, a_i(m_i) = \hat{a}_i(m_i) > 0$. Also, given Assumptions 1 and 2 and Equation 7, $u_i(0) = 0$. The derivative of Equation 7 for device $i$ is,

$$u_i'(m_i) = \phi_i'(a_i(m_i)) \cdot a_i'(m_i) - c_i. \tag{23}$$

**Case 1:** $\max_{m_i \geq 0} u_i(m_i) \leq 0$.

Each device $i$ starts with a utility of 0 since by Assumptions 1 and 2 $u_i(0) = 0$. Since $\max_{m_i \geq 0} u_i(m_i) \leq 0$, there is no utility gained by device $i$ to contribute more data. Therefore, the optimal amount of contributions remains at zero: $m_i^* = 0$.

**Case 2:** $\max_{m_i \geq 0} u_i(m_i) > 0$.

*Sub-Case 1:* $0 \leq m_i \leq m_0$. By definition of $m_0$, $a_i(m_i) = 0$ $\forall m_i \in [0, m_0]$. Therefore, from Equation 7 and Assumptions 1 and 2, $u_i(m_i) = -c_i m_i < 0$ $\forall m_i \in (0, m_0]$. Since $u_i(0) = 0$ and $u_i(m_i) < 0$ for $m_i > 0$, device $i$ will not collect any contribution: $m_i^* = 0$.

*Sub-Case 2:* $m_i > m_0$. Since $\forall m_i > m_0, a_i(m_i) = \hat{a}_i(m_i) > 0$, Equation 23 becomes,

$$u_i'(m_i) = \phi_i'(\hat{a}_i(m_i)) \cdot \hat{a}_i'(m_i) - c_i. \tag{24}$$

We begin by showing that $\phi_i(\hat{a}_i(m_i))$ is bounded. By Assumption 1, $\hat{a}_i(m_i) < a_{opt}^i < 1$ $\forall m_i$. Thus, $\phi_i(\hat{a}_i(m_i)) < \phi_i(a_{opt}^i) < \infty$ $\forall m_i$ since $\hat{a}_i(m_i)$ and $\phi_i$ are non-decreasing and continuous by Assumptions 1 and 2. Due to $\phi_i(\hat{a}_i(m_i))$ being concave, non-decreasing, and bounded (by Assumption 2), we have from Equation 24 that the limit of its derivative,

$$\lim_{m_i \to \infty} \phi_i'(\hat{a}_i(m_i)) \cdot \hat{a}_i'(m_i) = 0 \implies \lim_{m_i \to \infty} u_i'(m_i) = -c_i < 0. \tag{25}$$

Since $\phi_i(\hat{a}_i(m_i))$ is concave and non-decreasing, its gradient $\phi_i'(\hat{a}_i(m_i)) \cdot \hat{a}_i'(m_i)$ is maximized when $m_i = m_0$ and is non-increasing afterwards. Using the maximal derivative location $m_i = m_0$ in union with Case 2 ($\max_{m_i \geq 0} u_i(m_i) > 0$) and Sub-Case 2 ($u_i(m) \leq 0$ $\forall m \in [0, m_0]$) yields $u_i'(m_0) > 0$ (the derivative must be positive in order to increase utility above 0).

Now that $u_i'(m_0) > 0$, $\lim_{m_i \to \infty} u_i'(m_i) < 0$, and $\phi_i'(\hat{a}_i(m_i)) \cdot \hat{a}_i'(m_i)$ is non-increasing, there must exist a maximum $m_i = m^*$ such that $\phi_i'(\hat{a}_i(m^*)) \cdot \hat{a}_i'(m^*) = c_i$. $\qquad\square$

**Corollary 1.** *For uniform accuracy-payoff functions & data distributions: $u_j(m_j^o) \geq u_k(m_k^o)$, $\phi_j(a_j(m)) = \phi_k(a_k(m))$, and $m_j^o \geq m_k^o$ if marginal costs satisfy $c_j \leq c_k$ $\forall j, k \in [n]$.*

*Proof.* With uniform payoff functions, each device $i$'s utility and utility derivative become

$$u_i(m_i) = \phi(a_i(m_i)) - c_i m_i, \quad u_i'(m_i) = \phi'(a_i(m_i)) \cdot a_i'(m_i) - c_i. \tag{26}$$

Due to $\phi_i(a_i(m))$ being concave and non-decreasing, its derivative $\phi'(a_i(m_i)) \cdot a'_i(m_i)$ is non-negative and non-increasing. Let $m_k^o$ be the optimal amount of data contribution by device $k$.

**Case 1:** $c_j = c_k$. By Equation 26, if $c_j = c_k$ then $0 = u'_k(m_k^o) = u'_j(m_k^o)$. This implies $m_k^o = m_j^o$ and subsequently $u_k(m_k^o) = u_j(m_k^o)$.

**Case 2:** $c_j < c_k$. By Equation 26, if $c_j < c_k$, then $0 = u'_k(m_k^o) < u'_j(m_k^o)$. Since $\phi(a_i(m_i))$ is concave and non-decreasing, its derivative is non-increasing with its limit going to 0. Therefore, more data $\epsilon > 0$ must be collected in order for $u'_j$ to reach zero (*i.e.* $u'_j(m_k^o + \epsilon) = 0$). This implies that $m_j^o = m_k^o + \epsilon > m_k^o$. Furthermore, since $0 = u'_k(m_k^o) < u'_j(m_k^o)$, the utility for device $j$ is still increasing at $m_k^o$ and is fully maximized at $m_j^o$. This implies that $u_j(m_j^o) > u_k(m_k^o)$. $\square$

**Theorem 3.** *Consider a device $i$ with marginal cost $c_i$ and accuracy payoff function $\phi_i$ satisfying Assumptions 2 and 3. Denote device $i$'s optimal local data contribution as $m_i^o$ and its subsequent accuracy $\bar{a}_i := a_i(m_i^o)$. Define the derivative of $\phi_i(a)$ with respect to $a$ as $\phi'_i(a)$. For any $\epsilon \to 0^+$ and marginal server reward $r(\boldsymbol{m}) \geq 0$, device $i$ has the following accuracy-shaping function $\gamma_i(m)$ for $m \geq m_i^o$,*

$$\gamma_i := \begin{cases} \frac{-\phi'_i(\bar{a}_i) + \sqrt{\phi'_i(\bar{a})^2 + 2\phi''_i(\bar{a}_i)(c_i - r(\boldsymbol{m}) + \epsilon)(m - m_i^o)}}{\phi''_i(\bar{a}_i)}, \\ \frac{(c_i - r(\boldsymbol{m}) + \epsilon)(m - m_i^o)}{\phi'_i(\bar{a}_i)} \quad \text{if } \phi''_i(\bar{a}_i) = 0 \end{cases} \tag{27}$$

*Given the defined $\gamma_i(m)$, the following inequality is satisfied for $m \in [m_i^o, m_i^*]$,*

$$\phi_i(\bar{a}_i + \gamma_i(m)) - \phi_i(\bar{a}_i) > (c_i - r(\boldsymbol{m}))(m - m_i^o). \tag{28}$$

*The new optimal contribution for each device $i$ becomes $m_i^* := \{m \geq m_i^o \mid a_C(m + \sum_{j \neq i} m_j) = \bar{a}_i + \gamma_i(m)\}$. Device $i$'s data contribution increases $m_i^* \geq m_i^o$ for any contribution $\boldsymbol{m}_{-i}$.*

*Proof.* By the mean value version of Taylor's theorem we have,

$$\phi_i(\bar{a} + \gamma_i) = \phi_i(\bar{a}) + \gamma_i \phi'_i(\bar{a}) + 1/2\gamma_i^2 \phi''_i(z), \quad \text{for some } z \in [\bar{a}, \bar{a} + \gamma_i]. \tag{29}$$

Since $\phi_i(a)$ is both increasing and convex with respect to $a$,

$$\phi_i(\bar{a} + \gamma_i) - \phi_i(\bar{a}) \geq \gamma_i \phi'_i(\bar{a}) + 1/2\gamma_i^2 \phi''_i(\bar{a}). \tag{30}$$

In order to ensure $\phi_i(\bar{a} + \gamma) - \phi_i(\bar{a}) > (c_i - r(\boldsymbol{m}))(m - m_i^o)$, we must select $\gamma$ such that,

$$\gamma_i \phi'_i(\bar{a}) + 1/2\gamma_i^2 \phi''_i(\bar{a}) > (c_i - r(\boldsymbol{m}))(m - m_i^o). \tag{31}$$

**Case 1:** $\phi''_i(\bar{a}) = 0$. In this case, Equation 31 becomes,

$$\gamma_i \phi'_i(\bar{a}) > (c_i - r(\boldsymbol{m}))(m - m_i^o). \tag{32}$$

In order for Equation 31, and thereby Equation 12, to hold we select $\epsilon \to 0^+$ such that,

$$\gamma_i := \frac{(c_i - r(\boldsymbol{m}) + \epsilon)(m - m_i^o)}{\phi'_i(\bar{a})}. \tag{33}$$

**Case 2:** $\phi''_i(\bar{a}) > 0$. Determining when the left- and right-hand sides of Equation 31 are equal is equivalent to solving the quadratic equation for $\gamma_i$,

$$\gamma_i = \frac{-\phi'_i(\bar{a}) \pm \sqrt{\phi'_i(\bar{a})^2 + 2\phi''_i(\bar{a})(c_i - r(\boldsymbol{m}))(m - m_i^o)}}{\phi''_i(\bar{a})} \tag{34}$$

$$= \frac{-\phi'_i(\bar{a}) + \sqrt{\phi'_i(\bar{a})^2 + 2\phi''_i(\bar{a})(c_i - r(\boldsymbol{m}))(m - m_i^o)}}{\phi''_i(\bar{a})}. \tag{35}$$

The second equality follows from $\gamma_i$ having to be positive. In order for Equation 31, and thereby Equation 12, to hold we select $\epsilon \to 0^+$ such that,

$$\gamma_i := \frac{-\phi'_i(\bar{a}) + \sqrt{\phi'_i(\bar{a})^2 + 2\phi''_i(\bar{a})(c_i - r(\boldsymbol{m}) + \epsilon)(m - m_i^o)}}{\phi''_i(\bar{a})} \tag{36}$$

As a quick note, for $m = m_i^o$ one can immediately see that $\gamma_i(m) = 0$. To finish the proof, now that Equation 12 is proven to hold for the prescribed $\gamma_i$, device $i$ is incentivized to contribute more as the added utility $\phi_i(\bar{a} + \gamma_i) - \phi_i(\bar{a})$ is larger than the incurred cost $(c_i - r(\boldsymbol{m}))(m - m_i^o)$. There is a limit to this incentive, however. The maximum value that $\gamma_i$ can be is bounded by the accuracy from all contributions: $a_i(m_i^o) + \gamma_i \leq a_C(\sum_j m_j)$. The existence of the bound is guaranteed by Assumption 3. Thus, device $i$ reaches a new optimal contribution $m_i^*$ which is determined by,

$$m_i^* := \{m \geq m_i^o \mid a_C(m + \sum_{j \neq i} m_j) = a_i(m_i^o) + \gamma_i(m)\} \geq m_i^o. \tag{37}$$

$\square$

**Theorem 4.** REALFM $\mathcal{M}_R$ *(Equation 9) performs accuracy-shaping with $\gamma_i$ defined in Theorem 3 for each device $i \in [n]$ and some $\epsilon \to 0^+$. As such, $\mathcal{M}_R$ is Individually Rational (IR) and has a unique Nash equilibrium at which device $i$ will contribute $m_i^* \geq m_i^o$ updates, thereby eliminating the free-rider phenomena. Furthermore, since $\mathcal{M}_R$ is IR, devices are incentivized to participate as they gain equal to or more utility than by not participating.*

*Proof.* We first prove existence of a unique Nash equilibrium by showcasing how our mechanism $\mathcal{M}_R$ fulfills the criteria laid out in Theorem 1. The criteria in Theorem 1 largely surrounds the utility of a participating device $i$,

$$u_i^r(m_i; \boldsymbol{m}_{-i}) := [\mathcal{M}_R^U(m_i; \boldsymbol{m}_{-i})]_i - c_i m_i. \tag{38}$$

**Feasibility.** Before beginning, we note that $\mathcal{M}_R$ trivially satisfies the only non-utility requirement that $[\mathcal{M}_R^U(0; \boldsymbol{m}_{-i})]_i = 0$ (as $a_i(0) = \phi_i(0) = 0$). As shown in Equation 9, $\mathcal{M}_R$ returns accuracies between 0 and $a_i(\sum \boldsymbol{m})$ to all devices. This satisfies the bounded accuracy requirement. Furthermore, the utility provided by our mechanism $\mathcal{M}_R^U$ is bounded as well. Since $a_{opt}$ is the largest attained accuracy by our defined accuracy function $\hat{a}_i(m)$ and $a_{opt} < 1$, the maximum utility is $\phi_i(a_{opt}) < \infty$. Now, that $\mathcal{M}_R$ is proven to be Feasible, we only need the following to prove that $\mathcal{M}_R$ has a pure equilibrium: **(1)** $u_i^r(m_i; \boldsymbol{m}_{-i})$ is continuous in $\boldsymbol{m}_{-i}$ and **(2)** quasi-concave for $m_i \geq m_i^u := \inf\{m_i \mid [\mathcal{M}_R^U(m_i; \boldsymbol{m}_{-i})]_i > 0\}$.

**Continuity.** By definition of $u_i^r(m_i; \boldsymbol{m}_{-i})$, Equation 38, we only need to consider $[\mathcal{M}_R^U(m_i; \boldsymbol{m}_{-i})]_i$ since that is the only portion affected by $\boldsymbol{m}_{-i}$. By definition of the utility returned by our mechanism $\mathcal{M}_R$, shown in Equation 10, no discontinuities arise for a fixed $m_i$ and varying $\boldsymbol{m}_{-i}$. By assumptions on continuity in Assumptions 1 and 2, $\phi_i(a_i(m))$ is continuous for all $m$. Thus, for non-zero utility (zero utility would lead to zero reward), we find the marginal monetary reward function $r(\boldsymbol{m})$ in Equation 13 is continuous. Therefore, each piecewise component of $[\mathcal{M}_R^U(m_i; \boldsymbol{m}_{-i})]_i$ is continuous since they are sums of continuous functions. Finally, we show that the piecewise functions connect with each other continuously. The accuracy-shaping function $\gamma_i$ is defined such that $\gamma_i(m_i^o) = 0$ and $a_i(m_i^o) + \gamma_i(m_i^*) = a_i(\sum \boldsymbol{m})$, which finishes proof of continuity.

**Quasi-Concavity.** For all values of $m_i \geq m_i^u$, our mechanism $\mathcal{M}_R$ produces positive utility. By construction, our mechanism $\mathcal{M}_R$ is strictly increasing for $m_i \geq m_i^u$. Our mechanism $\mathcal{M}_R$ returns varying utilities within three separate intervals. While piece-wise, these intervals are continuous and $\mathcal{M}_R$ is strictly increasing with respect to $m_i$ in each. The first interval, consisting of the concave function $\phi_i(a_i(m_i))$, is quasi-concave by construction. The second interval consists of a linear function $r(\boldsymbol{m}) \cdot (m_i - m_i^o)$ added to a quasi-concave $\phi_i(a_i(m_i) + \gamma_i(m_i))$ function, resulting in a quasi-concave function (note that $\phi_i(\hat{a}_i(m_i))$ is concave). Finally, the third interval consists of a linear function $r(\boldsymbol{m}) \cdot (m_i - m_i^*)$ added to a concave function $\phi_i(a_i(m_i))$, which is also quasi-concave. In sum, this makes $[\mathcal{M}_R^U(m_i; \boldsymbol{m}_{-i})]_i$ a quasi-concave function. Since $-c_i m_i$ is a linear function, the utility of a participating device $u_i^r(m_i; \boldsymbol{m}_{-i})$ will also be quasi-concave function, as the sum of a linear and quasi-concave function is quasi-concave.

**Existence of Pure Equilibrium with Increased Data Contribution.** Since $[\mathcal{M}_R^U(\boldsymbol{m})]_i$ satisfies feasibility, continuity, and quasi-concavity requirements, $\mathcal{M}_R$ is guaranteed to have a pure Nash equilibrium by Theorem 1. Furthermore, since $\mathcal{M}_R$ performs accuracy-shaping with $\gamma_i$ prescribed in Theorem 1, it is guaranteed that each device $i$ will produce $m_i^* \geq m_i^o$ updates.

**Individually Rational (IR).** We prove $\mathcal{M}_R$ is IR by looking at each piecewise portion of Equation 9:

*Case 1: $m_i \leq m_i^o$ (Free-Riding).* When $m_i \leq m_i^o$, a device would attempt to provide as much or less than the amount of contribution which is locally optimal. The hope for such strategy would be free-riding: enjoy the performance of a well-trained model as a result of federated training while providing few (or zero) data points in order to save costs. Our mechanism avoids the free rider problem trivially by returning a model with an accuracy that is proportional to the amount of data contributed by the device. This is shown in Equation 9, as devices receive a model with accuracy $a_i(m_i)$ if $m_i \leq m_i^o$ (*i.e.,* devices are rewarded with a model equivalent to one that they could've trained themselves if they fail to contribute an adequate amount of data). In this case, devices receive the same model accuracy as they would've on their own and thus IR is satisfied in this case.

*Case 2: $m_i \in (m_i^o, m_i^*]$.* Via the results of Theorem 3, the accuracy of the model returned by $\mathcal{M}_R$ when $m_i \in (m_i^o, m_i^*]$ is greater than a model trained by device $i$ on $m_i$ local contributions. Mathematically, this is described as $a^r(m_i) = a_i(m_i) + \gamma_i(m_i) > a_i(m_i)$ for $m_i \in (m_i^o, m_i^*]$. Since $\phi_i$ is increasing, IR must hold as accuracy from $\mathcal{M}_R$ outstrips local training accuracy.

*Case 3: $m_i \geq m_i^*$.* By Theorem 3, by definition of $m_i^*$ when $m_i = m_i^*$ then the accuracy of a returned model by $\mathcal{M}_R$ is equal to $a_i(\sum_j m_j)$. Therefore, given a fixed set of contributions from all other devices $\boldsymbol{m_{-i}}$, device $i$ will still attain a model with accuracy $a_i(\sum_j m_j)$ for $m_i \geq m_i^*$ (since the limits of accuracy shaping have been reached for the given contributions). Due to this, IR trivially holds as $a_i(\sum_j m_j) \geq a_i(m_i)$. $\qquad\square$

## C.1 ACCURACY MODELING

Our model for accuracy stems from Example 2.1 in Karimireddy et al. (2022), which in turn comes from Theorem 11.8 in Mohri et al. (2018). Below is the mentioned generalization bound,

**Proposition 1** (Generalization Bounds, Karimireddy et al. (2022) Example 2.1). *Suppose we want to learn a model $h$ from a hypothesis class $\mathcal{H}$ which minimizes the error over data distribution $\mathcal{D}$, defined to be $R(h) := \mathbb{E}_{(x,y)\sim\mathcal{D}}[e(h(x), y)]$, for some error function $e(\cdot) \in [0, 1]$. Let such an optimal model have error $(1 - a_{opt}) \leq 1$. Now, given access to $\{(x_l, y_l)\}l \in [m]$ which are $m$ i.i.d. samples from $\mathcal{D}$, we can compute the empirical risk minimizer (ERM) as $\hat{h}_m = \arg\min_{h\in\mathcal{H}} \sum_{l\in[m]} e(h(x), y)$. Finally, let $k > 0$ be the pseudo-dimension of the set of functions $\{(x, y) \to e(h(x), y) : h \in \mathcal{H}\}$, which is a measure of the difficulty of the learning task. Then, standard generalization bounds imply that with probability at least 99% over the sampling of the data, the accuracy is at least*

$$1 - R(\hat{h}_m) \geq \left\{ \hat{a}(m) := a_{opt} - \frac{\sqrt{2k(2 + \log(m/k))} + 4}{\sqrt{m}} \right\}. \tag{39}$$

*A simplified expression for our analytic analysis use is,*

$$\hat{a}(m) = a_{opt} - 2\sqrt{k/m}. \tag{40}$$

