# OpenReview forum: "Towards Realistic Mechanisms That Incentivize Federated Participation and Contribution"
_ICLR.cc/2025/Conference — Submitted to ICLR 2025_

### Official Review · Reviewer_uxm7 · 2024-11-01

**Soundness:** 3
**Presentation:** 2
**Contribution:** 2
**Rating:** 5
**Confidence:** 3

**Summary:**

This work proposes a federated learning mechanism to incentivize data contribution and device participation. Building on the work of [1], this paper further allows more realistic settings such as non-linear mapping between utility and model accuracy, non-iid local data distribution (to some extent), preventing data sharing, and modeling of server utility.

**Strengths:**

- The mapping from the model accuracy to utility is non-linear, which renders it more realistic.

- Server utility is explicitly modeled, which has not been widely explored in previous research.

**Weaknesses:**

- In calculating server accuracy, it’s unclear why Assumption 3 holds, especially if local accuracies vary significantly across devices.

- The study aims to address a cross-device setup; however, the experiments involve only 16 devices, which somewhat limits the persuasiveness of the empirical results.

- The comparison of utilities in Figure 3 seems unfair. For the non-linear and linear methods, agents’ utilities are distinct functions of model accuracy, meaning that even with the same model accuracy, their utilities would differ.

- Typo line 412: incentives -> incentivizes

**Questions:**

- The authors claim that the work does not involve data sharing, but $c_im_i$ formulation is the same as [1]. Does this imply that the analysis from [1] can readily be extended to cases without data sharing?

- In Figure 4, data contribution from non-linear RealFM in MNIST case goes to 10^8 magnitude, while it is 10^4 for Cifar10 case. I find this unreasonable, as Cifar10 is a harder task and more data will be helpful. This has something to do with the chosen phi function, where utility can go to infinity when accuracy is close to 1 (which is the case for easier tasks).

[1] Karimireddy et al. Mechanisms that Incentivize Data Sharing in Federated Learning

---

> ### Author Response · Authors · 2024-11-19
> **Reviewer uxm7 Rebuttal**
>
> Thank you, Reviewer uxm7, for your insightful review of our paper. Below, we address all questions you raised.
>
> ---
>
> ## Weaknesses
>
> > **Weakness 1:** In calculating server accuracy, it’s unclear why Assumption 3 holds, especially if local accuracies vary significantly across devices.
>
> **Response to Weakness 1:**
>
> - Assumption 3 holds the majority of the time, since the server is able to train on much more data than any single device.
> - We note in our paper that the assumptions in RealFM may be violated in certain non-i.i.d settings, however empirically we find that RealFM performs well on highly-skewed datasets.
> - RealFM is the first federated mechanism to eliminate federated free riding while simultaneously incentivizing increased participation and contribution *outside of the i.i.d setting*.
>
>
> > **Weakness 2:** The study aims to address a cross-device setup; however, the experiments involve only 16 devices, which somewhat limits the persuasiveness of the empirical results.
>
> **Response to Weakness 2:**
>
> - We provide experimental results for both 8 and 16 devices in our paper.
> - Our experiments are truly distributed, using MPI. As a result, we do not have the computational resources to scale up to more devices.
>
> Unlike other FL papers, which only simulate the FL process, we create a codebase to perform truly parallel federated training. However, due to a lack of computational resources (GPUs), we were unable to scale to more than 16 devices. We expect RealFM to scale well, as it is a reward-payment mechanism that has little computational overhead. As a result, it can be used in tandem with various efficient and scalable federated training methods.
>
>
> > **Weakness 3:** The comparison of utilities in Figure 3 seems unfair. For the non-linear and linear methods, agents’ utilities are distinct functions of model accuracy, meaning that even with the same model accuracy, their utilities would differ.
>
> **Response to Weakness 3:**
>
> - Utilizing a linear relationship between accuracy and utility is unrealistic and consequently reduces the optimal amount of data each device uses for training (which also reduces the final model performance).
>
> We agree with the reviewer that the utilities are distinct functions of model accuracy. In Figure 3, the linear version of RealFM does have reduced server utility due to the usage of an unrealistic relationship between accuracy and utility. However, there also is a reduction in server utility due to the fact that devices using a linear relationship will train on less data. This fact arises from Theorem 2, as a convex and increasing $\phi$ spurs a larger optimal data amount $m^*$. Since devices train on less data when using a linear relationship, the resulting model will be less accurate than when using a non-linear relationship.
>
> ---
>
> ## Questions
>
> > **Question 1:** The authors claim that the work does not involve data sharing, but $c_im_i$ formulation is the same as [1]. Does this imply that the analysis from [1] can readily be extended to cases without data sharing?
>
> **Response to Question 1:**
>
> - Our formulation is not the same as [1]. We utilize an accuracy-payoff function $\phi$ that can be non-linear, thereby vastly changing our analysis.
> - The term $c_im_i$ term simply quantifies the cost of data collection and training. It does not signify data sharing.
> - The analysis in [1] utilizes an i.i.d setting only, which holds within the data sharing domain yet rarely holds in federated domains.
>
> > **Question 2:** In Figure 4, data contribution from non-linear RealFM in MNIST case goes to 10^8 magnitude, while it is 10^4 for Cifar10 case. I find this unreasonable, as Cifar10 is a harder task and more data will be helpful. This has something to do with the chosen phi function, where utility can go to infinity when accuracy is close to 1 (which is the case for easier tasks).
>
>
> **Response to Question 2:**
>
> - It is highly desirable to attain 100% accuracy, even for easier learning tasks.
>
> It is realistic for devices to be incentivized to train a model to be near 100% accurate. In settings like healthcare, many people would be highly incentivized to contribute more to attain a few percentage points increase in accuracy.
>
> - The accuracy-payoff function $\phi$ is user and domain specific. For easier tasks, it can become closer towards a linear function while it can be highly non-linear for harder ones.
>
> In our MNIST experiments, $\phi$ likely should be somewhere between the linear RealFM and non-linear RealFM $\phi$ values used. We will perform additional studies for varying $\phi$ values on MNIST.

---

> > ### Comment · Reviewer_uxm7 · 2024-11-25
> >
> > Thank you for your response.
> >
> > Regarding W3, I believe it would be better to revise Figure 3 in the manuscript to show a plot of the final model accuracy, as you suggested. The current presentation makes it difficult to ensure a fair comparison.
> >
> > As for Q1, based on your clarification, the term $c_i m_i$ can be interpreted in various ways. I believe this suggests that the work by Karimireddy et al. could also be naturally extended to a non-data-sharing setup.
> >
> > I will keep my score as it was.

---

> > > ### Author Response · Authors · 2024-11-25
> > > **Discussion Response**
> > >
> > > Dear Reviewer uxm7,
> > >
> > > Thank you for your response. Below are our responses to your follow-up comments.
> > >
> > > > **Comment:** Regarding W3, I believe it would be better to revise Figure 3 in the manuscript to show a plot of the final model accuracy, as you suggested. The current presentation makes it difficult to ensure a fair comparison.
> > >
> > > **Response:**
> > >
> > > As mentioned within our rebuttal to W3, we note that the non-linear method "spurs a larger optimal data amount $m^*$". Thus, we utilize Figure 4 to showcase this large improvement in data usage. Figure 3 and 4 are inherently tied together, and we will try to pair them within our paper to better emphasize this.
> > >
> > >
> > > > **Comment:** As for Q1, based on your clarification, the term $c_im_i$ can be interpreted in various ways. I believe this suggests that the work by Karimireddy et al. could also be naturally extended to a non-data-sharing setup.
> > >
> > > **Response:**
> > >
> > > - Karimireddy et al. paints itself as a data-sharing method because of the assumptions it utilizes (the name of their paper is "Mechanisms that Incentivize **Data Sharing** in Federated Learning").
> > >
> > > As we mention within our paper, as well as our original rebuttal, that our mechanism relaxes previous unrealistic assumptions by allowing:
> > > 1. A non-linear relationship between accuracy and utility,
> > > 2. Non-iid data settings,
> > > 3. Modeling of the central server utility.
> > >
> > > Without these improvements, it is not possible to easily extend Karimireddy's work to non-data-sharing setups. **That is the reason why they market their paper as a data sharing mechanism.**
> > >
> > > - In contrast to Karimireddy et al, our paper explicitly details a truly FL algorithm (Algorithm 1).

---

### Official Review · Reviewer_E9ZQ · 2024-11-03

**Soundness:** 3
**Presentation:** 3
**Contribution:** 2
**Rating:** 5
**Confidence:** 4

**Summary:**

The paper introduces an incentive mechanism for federated learning, addressing key issues like the free-rider problem and the lack of realistic incentives. The approach is well-motivated, theoretically solid, and supported by experiments demonstrating improvements over existing methods. However, some areas, such as more comprehensive literature reviews in related work, practical implementation details, and additional ablation studies, could further strengthen the work.

**Strengths:**

1. Well-written: The paper is clear and well-structured.
2. Solid Theoretical Analysis: The theoretical analysis and proofs are strong and well-supported.

**Weaknesses:**

1.	Free-rider problem has been widely studied in FL settings, such as [1], and literature in its related works. In addition, previous studies of mechanisms for FL-related scenarios, such as crowdsourcing, can be suitable and easily adapted to FL settings.
2.	Achieving the goals of this paper seems to require knowledge of the data amount on each device. However, in the context of privacy-preserving machine learning, is it essential for FL clients to disclose their data sizes? This raises the question of whether the proposed approach truly aligns with the notion of “REALISTIC MECHANISMS” as suggested in the title.
3. Including well-organized source code for the experiments would enhance the paper's reproducibility and allow reviewers to verify specific details.

[1] Meng J, et al. Federated Learning and Free-riding in a Competitive Market, 2024

**Questions:**

1. Are some ablation studies missing, such as examining performance under different client numbers or aggregation algorithms? （A mini Question)

---

> ### Author Response · Authors · 2024-11-19
> **Reviewer E9ZQ Rebuttal**
>
> Thank you, Reviewer E9ZQ, for your insightful review of our paper. Below, we address all questions you raised.
>
> ---
>
> ## Weaknesses
>
> > **Weakness 1:** Free-rider problem has been widely studied in FL settings, such as [1], and literature in its related works. In addition, previous studies of mechanisms for FL-related scenarios, such as crowdsourcing, can be suitable and easily adapted to FL settings.
>
> **Response to Weakness 1:**
>
> We first want to thank the reviewer for bringing to our attention this recent and very interesting paper. We will add discussion of our paper versus [1] within our related works section.
>
> - Unlike the majority of free-rider work within FL, including crowdsourcing works, RealFM provably incentivizes increased device participation and data contribution.
>
> RealFM provides theoretical Nash guarantees that its usage will result in not just the elimination of free riding, but improved participation and contribution towards federated training!
>
> > **Weakness 2:** Achieving the goals of this paper seems to require knowledge of the data amount on each device. However, in the context of privacy-preserving machine learning, is it essential for FL clients to disclose their data sizes? This raises the question of whether the proposed approach truly aligns with the notion of “REALISTIC MECHANISMS” as suggested in the title.
>
> **Response to Weakness 2:**
>
> - Traditional federated learning methods utilize a weighted averaging scheme that is proportional to the size of each device's data.
>
> This is a standard approach in FL, and **this also assumes that the server knows the size of each device's data**.
>
> - Providing the dataset size to a server does not leak either any of the data itself or any characteristics of the data (*e.g.* its distribution).
>
>
> > **Weakness 3:** Including well-organized source code for the experiments would enhance the paper's reproducibility and allow reviewers to verify specific details.
>
> **Response to Weakness 3:**
>
> We have provided source code with a readme, however we will make the readme more concise and improve the overall organization of the code.
>
> ---
>
> ## Questions
>
> > **Question 1:** Are some ablation studies missing, such as examining performance under different client numbers or aggregation algorithms? （A mini Question)
>
> **Response to Question 1:**
>
> - We provide experimental results with 8 and 16 devices in our paper.
>
> Unlike other FL papers, which only simulate the FL process, we create a codebase to perform truly parallel federated training. However, due to a lack of computational resources (GPUs), we were unable to scale to more than 16 devices.
>
> - RealFM is a mechanism that proposes a novel, theory-backed reward payment system that can be used in conjunction with various aggregation algorithms.
>
> Since RealFM is agnostic to the aggregation algorithm, an ablation study across various aggregation algorithms does not provide any additional insight into RealFM.

---

> > ### Comment · Reviewer_E9ZQ · 2024-11-23
> >
> > **Response to your answer to my questions**
> >
> > Thank you for your reply! I understand the limitations regarding source availability for running on more devices. However, I was wondering why you specifically chose 8 and 16 devices, rather than other configurations like 10, 12, or 14.
> > I am curious to know whether your method would also achieve good performance under these alternative settings.
> >
> > Thank you for your time and clarification!

---

> > ### Comment · Reviewer_E9ZQ · 2024-11-23
> >
> > **Response to Weakness 1**
> >
> > Thank you for taking the time to respond. There are several prior works addressing the free-rider problem in federated learning, such as the studies mentioned in the related work section of [1]. Instead of focusing solely on the literature I provided, I encourage you to conduct a more comprehensive investigation into this area. I look forward to seeing these additions in your revision.

---

> > > ### Author Response · Authors · 2024-11-25
> > > **Discussion Response**
> > >
> > > Dear Reviewer E9ZQ,
> > >
> > > Thank you for your response. Below are our responses to your follow-up comments.
> > >
> > > > **Comment:** However, I was wondering why you specifically chose 8 and 16 devices, rather than other configurations like 10, 12, or 14. I am curious to know whether your method would also achieve good performance under these alternative settings.
> > >
> > > **Response:**
> > >
> > > - We ran on 16 devices as this was the maximum number we were able to run on.
> > > - We wanted to scale by a factor of 2 so we also chose 8 devices.
> > > - **We are extremely confident that our performance holds for 10, 12, or 14 devices.**
> > >
> > > Our mechanism works extremely well for the number of devices both larger and smaller than 10, 12, or 14 devices. The small difference in the number of devices used will not cause any difference in the performance of our mechanism.
> > >
> > > > **Comment:** There are several prior works addressing the free-rider problem in federated learning, such as the studies mentioned in the related work section of [1]. Instead of focusing solely on the literature I provided, I encourage you to conduct a more comprehensive investigation into this area. I look forward to seeing these additions in your revision.
> > >
> > > **Response:**
> > >
> > > I believe that you may have missed the Related Works section, within Appendix A, that also cites many of the same works as in [1]. There is substantial overlap between the works cited in both papers. Nevertheless, we are able to include a few of the other cited papers within [1] as well.
> > >
> > > ---
> > >
> > > We are happy to address any remaining concerns if they exist. If all of your concerns have been addressed, we would request a reconsideration of the reviewer's original score.

---

> > > > ### Comment · Reviewer_E9ZQ · 2024-11-30
> > > >
> > > > Thank you for your response. I definitely have **carefully** read the Appendix and noted the Related Works section. However, I still find the literature review to lack comprehensiveness. The paper does not sufficiently explore or compare existing works in depth. Therefore, I will maintain my score.

---

> > > > > ### Author Response · Authors · 2024-12-02
> > > > >
> > > > > Dear Reviewer E9ZQ,
> > > > >
> > > > > Thank you for your comments. We are curious about what areas of our literature review need more comprehensiveness. Our paper details:
> > > > >
> > > > > 1. contract theory & FL free riding,
> > > > > 2. general federated mechanisms, and
> > > > > 3. collaborative learning and Shapley value approaches.
> > > > >
> > > > > In your original review, you mention comparing against crowdsourcing, which we already do within the Introduction and Federated Mechanism section of the Related Works and Appendix (Zhan 2020a;b; 2021) as well as within the Contract Theory section of the Related Works (Wang 2021). It is unclear what other specific areas we are missing, especially as there is already an overlap between what we cite and what is cited in the paper the reviewer mentions [1].
> > > > >
> > > > > Furthermore, as detailed in our paper and our rebuttal, the merits of our paper stem from our novel FL mechanism that provably incentivizes increased device participation and data contribution. This has not been accomplished before in literature, and we believe that this should carry the most weight when judging our paper.
> > > > >
> > > > > We are happy to add more literature if the reviewer specifically provides what research areas we are missing within our Related Works section.
> > > > >
> > > > > Best,
> > > > >
> > > > > Authors

---

### Official Review · Reviewer_PBT6 · 2024-11-04

**Soundness:** 2
**Presentation:** 3
**Contribution:** 2
**Rating:** 6
**Confidence:** 3

**Summary:**

The paper proposes REALFM, a federated learning mechanism designed to incentivize edge device participation and data contribution by modeling device utility and removing the free-rider dilemma under non-i.i.d distributions. It introduces non-linear reward structures to enhance both device and server utility in federated settings with heterogeneous data. The proposed approach shows significant improvements in device contributions and model accuracy compared to existing mechanisms.

**Strengths:**

-The paper provides a solid theoretical framework, supported by rigorous proofs and analyses
- The paper is well-organized, with clear definitions and explanations that make the methods and results easy to follow.

**Weaknesses:**

- The paper focuses on non-i.i.d case but uses "data heterogeneity" for so long before being precise, which can be misleading for the reader. Moreover, assumptions and mechanisms would struggle in highly variable data distributions: The accuracy shaping mechanism assumes that central server updates generally benefit all participants, which would not be the case for label or covariate shift.
- REALFM introduces a complex reward and incentive mechanism that may be challenging to implement in "real-world" federated learning systems. The added layers of utility modeling and accuracy shaping might make it difficult to scale across diverse device types and operational environments. Additionally, the "real" part might be an overestimation.
- It is unclear what kind of "realistic" scenarios can benefit from such incentive scheme.
- The figures could have better readability by adding more information about the datasets

**Questions:**

- Please explain why "for example, accuracy improvement from 48% to 49% should be rewarded much differently than 98% to 99%"
- Please explain how the proposed approach can be extended to covariate and concept shift.
- How is the proposed approach handling clients with minority data? The approach aims to punish free-riders but it might mistakenly punish minority clients, in particular by offering a noisy model
- Please add the dataset name and dirichlet parameter to the figures to improve readability.
- What kind of "real" applications would require this incentive scheme?

---

> ### Author Response · Authors · 2024-11-19
> **Reviewer PBT6 Rebuttal (Weaknesses)**
>
> Thank you, Reviewer PBT6, for your insightful review of our paper. Below, we address all questions you raised.
>
> ---
>
> ## Weaknesses
>
> > **Weakness 1:** The paper focuses on non-i.i.d case but uses "data heterogeneity" for so long before being precise, which can be misleading for the reader. Moreover, assumptions and mechanisms would struggle in highly variable data distributions: The accuracy shaping mechanism assumes that central server updates generally benefit all participants, which would not be the case for label or covariate shift.
>
> **Response to Weakness 1:**
> - RealFM is the first federated mechanism to eliminate federated free riding while simultaneously incentivizing increased participation and contribution *outside of the i.i.d setting*.
> - We note in our paper that the assumptions in RealFM may be violated in certain non-i.i.d settings, however empirically we find that RealFM performs well on highly-skewed datasets.
> - We will specify earlier within the paper about the type of data heterogeneity we consider in RealFM, and extending the theoretical results of RealFM to stricter non-i.i.d settings is an important line of future research.
>
> > **Weakness 2:** REALFM introduces a complex reward and incentive mechanism that may be challenging to implement in "real-world" federated learning systems. The added layers of utility modeling and accuracy shaping might make it difficult to scale across diverse device types and operational environments. Additionally, the "real" part might be an overestimation.
>
> **Response to Weakness 2:**
> - The addition of utility modeling and accuracy shaping requires only a small amount of computational overhead.
>
> As one can see in Algorithm 1, the server must loop over all participating devices and compute the rewards for each device. Computing the rewards only requires a few FLOPs. Empirically, the time to compute the rewards is tiny. It took only a few seconds (~2 seconds).
>
> - The "real" part of RealFM is the relaxation of assumptions versus previous federated mechanisms (as detailed in Sections 1 and 2).
>
> One example, related to this weakness, is the new, non-linear relationship between accuracy and utility. The rationale behind why this is much more realistic is provided in our response to Question 1 below.
>
> > **Weakness 3:** It is unclear what kind of "realistic" scenarios can benefit from such incentive scheme.
>
> **Response to Weakness 3:**
> Consider a realistic situation where a consortium of hospitals seek collaboration to train a model, privately in a FL manner, that can diagnose skin cancer. Now, let's say one of the hospitals is smaller and resource-constrained. It is difficult, but not impossible, for this hospital to collect more data for training. Thus, in the absence of a FL free-riding eliminating mechanism, the smaller hospital could contribute little to no data towards training while still reaping the rewards of a well-trained global model.
>
> RealFM provably ensures that it is in this smallest hospital's best interest to participate and not free ride (Theorem 4). Furthermore, it is actually best to contribute more data to FL training in order for this hospital to receive an improved model to diagnose skin cancer. **The overall effect is that all participating hospitals will want to contribute more data to training, which in turn will further improve the performance of the FL model.** This is especially important within the medical setting, where training data is hard to come by. Incentivizng hospitals to generate and contribute more data will improve ML performance within the medical field.
>
> > **Weakness 4:** The figures could have better readability by adding more information about the datasets.
>
> **Response to Weakness 4:**
> We will add more background information about the datasets into the figures for readers.

---

> > ### Comment · Reviewer_PBT6 · 2024-12-02
> >
> > Thank you for your response. I think the answer to the first two points are satisfactory. The answer about a realistic scenario makes me actually concerned. The goal is not just capitalistic, but should be public good of the patients. Giving a worse model that will be harmful to patients is not enough justification for this. Why would a hospital be malicious? Why should we punish a small community for not having enough data or resources to collect data with such mechanism. Please think of other examples and consider the possible harms of your approach!

---

> > > ### Author Response · Authors · 2024-12-02
> > > **Discussion Response**
> > >
> > > Dear Reviewer PBT6,
> > >
> > > We believe that there may be some confusion surrounding how our mechanism functions. In the example we describe, we do not punish any hospital for not having enough data or resources. Our mechanism is provably Individually Rational. **Thus, each participating hospital would only stands to gain benefit by participating in RealFM**. Our mechanism acts to incentivize all hospitals to actively participate in training and increase the amount of data to use during training. In no way will participating hospitals be worse off. On the contrary, they all will provably improve their utility. Participating hospitals provably receive models that are better than what they would have trained on their own.
> > >
> > > Another example is a company allowing users to collaboratively train a model for optimizing electricity usage throughout the day. Users receive a well-trained FL model in return for providing gradient updates (using their own data). In traditional FL settings, users could sign up to participate but contribute little to no gradient updates while still receiving the final model, which is unfair. In contrast, our mechanism provably incentivizes (1) more users to participate and (2) users to contribute more than what they would have on their own. Since users are incentivized to contribute more than what they would have on their own, there is no more risk of free-riding.
> > >
> > > Best,
> > >
> > > Authors

---

> ### Author Response · Authors · 2024-11-19
> **Reviewer PBT6 Rebuttal (Questions)**
>
> ## Questions
>
> > **Question 1:** Please explain why "for example, accuracy improvement from 48% to 49% should be rewarded much differently than 98% to 99%"
>
> **Response to Question 1:**
> Let's consider an example where a ML model is used to diagnose cancer in patients. A model with either 48% or 49% accuracy may be useful, but neither are reliable and dependable within healthcare settings. A model with 98% accuracy would be of great use by doctors, however there would still be unfortunate and life-altering misdiagnoses. We surmise that the ability to classify 99% of cancer diagnoses, thereby reducing the number of misdiagnoses, would be highly sought after by doctors and the medical community at large.
>
>
> > **Question 2:** Please explain how the proposed approach can be extended to covariate and concept shift.
>
> **Response to Question 2:**
>
> - RealFM is flexible enough to leverage various FL methods that alleviate covariate [Tan 2024] and concept [Jothimurugesan 2023] shifts.
>
> RealFM is training-method agnostic. The mechanism we present solely determines how to adequately reward participating agents such that they will provably participate and contribute more to training and not free ride. Thus, RealFM allows various FL techniques to fight the issues of covariate and concept shift.
>
> - RealFM is the first federated mechanism in its area that provides guarantees *outside of the i.i.d setting*.
> - Empirically, we find that RealFM performs well on highly-skewed datasets (where label shift occurs).
>
> > **Question 3:** How is the proposed approach handling clients with minority data? The approach aims to punish free-riders but it might mistakenly punish minority clients, in particular by offering a noisy model.
>
> **Response to Question 3:**
>
>
> - Devices with minority data have large data costs $c_i$ (as it is expensive to collect this data).
> - Our accuracy-shaping mechanism (Theorem 3) proves that even devices with large costs are incentivized to contribute more to training, and as a result will receive the well-trained global model.
>
> We prove in Theorem 3 that devices with minority data, and thus large data costs, will still receive the global model if they contribute even a few more data points towards training (Equation 12). As a result, these devices will neither (i) be forced to contribute an amount of data that is too costly or (ii) receive a noisy model
>
> - Incentivizing agents with minority data, that also have extremely high costs, to contribute and possibly collect more data is an important future line of research.
>
>
> > **Question 4:** Please add the dataset name and dirichlet parameter to the figures to improve readability.
>
> **Response to Question 4:**
> We thank the reviewer for their suggestion and will update this within our paper.
>
> > **Question 5:** What kind of "real" applications would require this incentive scheme?
>
> We provide an example application in our response to Weakness 3.
>
>
> 1. Tan, Yue, et al. "Is heterogeneity notorious? taming heterogeneity to handle test-time shift in federated learning." Advances in Neural Information Processing Systems, 2024.
> 2. Jothimurugesan, Ellango, et al. "Federated learning under distributed concept drift." International Conference on Artificial Intelligence and Statistics. PMLR, 2023.

---

### Official Review · Reviewer_yGmp · 2024-11-06

**Soundness:** 3
**Presentation:** 2
**Contribution:** 2
**Rating:** 5
**Confidence:** 3

**Summary:**

The paper introduces incentive mechanisms in federated learning (FL).

**Strengths:**

REALFM incentivizes edge devices to participate in federated learning by offering accuracy-based and monetary rewards proportional to each device's data contribution. This mechanism enables the central server to attain better model performance by motivating devices to contribute more data than they would on their own.

**Weaknesses:**

The paper lacks a detailed discussion on the computational overhead of implementing REALFM, especially given the added accuracy-shaping and monetary reward calculations. Including complexity analysis would help readers understand potential trade-offs.

The paper could be enhanced if  it could further exploring the impact of varying device capabilities (e.g., computational power, storage) on REALFM’s effectiveness and scalability.

A discussion on privacy-preserving mechanisms to protect device-specific information in the proposed setups would enhance the paper’s relevance to practical federated applications.

Given that contract-based FL mechanisms (e.g., using registration fees to penalize free-riders) are common in the literature, adding a comparative analysis with these mechanisms would offer a more rounded assessment of REALFM’s advantages and limitations.

**Questions:**

The paper lacks a detailed discussion on the computational overhead of implementing REALFM, especially given the added accuracy-shaping and monetary reward calculations. Including complexity analysis would help readers understand potential trade-offs.

The paper could be enhanced if  it could further exploring the impact of varying device capabilities (e.g., computational power, storage) on REALFM’s effectiveness and scalability.

A discussion on privacy-preserving mechanisms to protect device-specific information in the proposed setups would enhance the paper’s relevance to practical federated applications.

Given that contract-based FL mechanisms (e.g., using registration fees to penalize free-riders) are common in the literature, adding a comparative analysis with these mechanisms would offer a more rounded assessment of REALFM’s advantages and limitations.

---

> ### Author Response · Authors · 2024-11-19
> **Reviewer yGmp Rebuttal**
>
> Thank you, Reviewer yGmp, for your insightful review of our paper. Below, we address all questions you raised.
>
> ---
>
> ## Weaknesses
>
> > **Weakness 1:** The paper lacks a detailed discussion on the computational overhead of implementing REALFM, especially given the added accuracy-shaping and monetary reward calculations. Including complexity analysis would help readers understand potential trade-offs.
>
> **Response to Weakness 1:**
>
> - RealFM has minimal added computational overhead, only requiring simple calculations of $m_i^o, m_i^*, a_i^r, R_i$ by the server for each device $i$.
> - The number of computations (all occurring post-training) scales with the number of participating devices.
>
> As one can see in Algorithm 1, the server must loop over all participating devices and compute the rewards for each device. Computing the rewards only requires a few FLOPs. Empirically, the time to compute the rewards is tiny. It took only a few seconds (~2 seconds).
>
> > **Weakness 2:** The paper could be enhanced if it could further exploring the impact of varying device capabilities (e.g., computational power, storage) on REALFM’s effectiveness and scalability.
>
> **Response to Weakness 2:**
>
> - RealFM is a mechanism that can be added on top of *any* federated training method.
> - The only added cost incurred by using RealFM is the small server computational costs (detailed in Weakness 1 response) and one-time communication costs when the server provides each device with their rewards.
>
> RealFM does not affect the actual federated learning process. For example, RealFM can be implemented on top of an asynchronous or compressed federated learning method without issue. This flexibility is a major strength of RealFM.
>
> > **Weakness 3:** A discussion on privacy-preserving mechanisms to protect device-specific information in the proposed setups would enhance the paper’s relevance to practical federated applications.
>
> **Response to Weakness 3:**
>
> - The only device-specific information required by RealFM are the device costs, utility, and optimal accuracy.
>
> No actual device data is required to be sent by any devices to the server in RealFM. Furthermore, as mentioned in the response to Weakness 2, privacy-preserving federated learning methods can be enacted in conjunction with RealFM to reduce privacy issues.
>
> - Allowing noisy device costs, utility, and optimal accuracy values is an active line of follow-up research we are pursuing.
>
> The goal of our work is to provide a federated mechanism that correctly models the relationship between accuracy and utility and provably incentivizes devices to participate and contribute more than they would locally on their own. We hope to build on top of this in future work by allowing noisy device information and still reaching an equilibrium.
>
> > **Weakness 4:** Given that contract-based FL mechanisms (e.g., using registration fees to penalize free-riders) are common in the literature, adding a comparative analysis with these mechanisms would offer a more rounded assessment of REALFM’s advantages and limitations.
>
> **Response to Weakness 4:**
> - We compare existing contract-based FL mechanism to RealFM within the Related Works (Section 2).
> - Existing contract-based methods focus on optimal contract and reward design, whereas our work designs an incentive scheme.
> - Contract-based FL mechanisms penalize free-riders yet do not provide incentives for devices to participate and contribute more data.
>
> ---
>
> ## Questions
>
> We believe that our response to the weaknesses above will answer the questions listed by the reviewer.

---

> ### Comment · Reviewer_yGmp · 2024-12-02
>
> Thank you for your response. The response has partially addressed my concerns, but I still have concerns regarding weakness 4. Thus I will only raise it to 5.

---

> > ### Author Response · Authors · 2024-12-02
> > **Discussion Response**
> >
> > Dear Reviewer yGmp,
> >
> > Thank you for your response. We are curious about what of your concerns remain for Weakness 4.
> >
> > > **Weakness 4**: Given that contract-based FL mechanisms (e.g., using registration fees to penalize free-riders) are common in the literature, adding a comparative analysis with these mechanisms would offer a more rounded assessment of REALFM’s advantages and limitations.
> >
> > - We compare existing contract-based FL mechanism to RealFM within the Related Works (Section 2).
> >
> > Are there other specific mechanisms that you feel that we left out? We detail more than five prominent FL contract theory papers, and explicitly detail the differences between them and our work. Namely that,
> >
> > 1. Existing contract-based methods focus on optimal contract and reward design, whereas our work designs an incentive scheme.
> > 2. Contract-based FL mechanisms penalize free-riders yet do not provide incentives for devices to participate and contribute more data.
> >
> > **Remark:**
> >
> > - The merits of our paper stem from our novel FL mechanism that provably incentivizes increased device participation and data contribution.
> >  - Our technical contributions have not been accomplished before in literature, and we believe that this should carry the most weight when judging our paper.
> >
> > We are happy to add more literature if the reviewer specifically provides what research areas we are missing within our Related Works section.
> >
> > Best,
> >
> > Authors

---

### Meta-Review · Area_Chair_iSuh · 2024-12-20

**Metareview:**

a) Summary

This work introduces a novel federated learning mechanism designed to address the free-rider problem by incentivizing data contribution and device participation while maintaining individual rationality. It incorporates a non-linear relationship between model accuracy and utility, allowing more realistic modeling of device and server utility compared to prior works. The mechanism provides theoretical guarantees for eliminating free-riding and improving data contribution and model performance in non-i.i.d. settings.

b) Strengths
- Important Problem Addressed: The paper tackles the critical issue of the free-rider problem in federated learning and attempts to provide a theoretical framework to understand and mitigate the issue.
- More realistic utilities: The paper introduces a non-linear relationship between model accuracy and utility, which more closely reflects real-world applications compared to existing linear approaches.
- Experimental evaluations: Significant improvements (3-4 magnitudes) in data contribution and server utility are demonstrated on real-world datasets.

c) Weakness
- Ambiguous "Realistic" Claims: The paper's claim of being "realistic" is weakened by reliance on assumptions like Assumption 3, which may not hold in highly heterogeneous or non-i.i.d. settings. Practical applicability to real-world FL systems remains unclear.
- Lack of theoretical novelty: While the authors claim that [Karimireddy et al. 20]'s work cannot be extended to non-data-sharing scenarios due to its assumptions, both the reviewers and I remain unconvinced. Their proofs nearly directly translate to the results here. In fact, they are even simpler since the authors assume concave utilities (which was not the case in Karimireddy et al.).
- Literature review: The reviewers remain unconvinced the authors did a thorough enough job of reviewing the related literature.

d) Reason for **rejection**

The paper's claim of being "realistic" is undermined by reliance on fragile assumptions, like Assumption 3, which may not hold in highly heterogeneous or non-i.i.d. settings, making its practical applicability unclear. Theoretical novelty is also lacking, as many of the results appear to directly translate from prior work by Karimireddy et al. Additionally, the literature review remains insufficient, with reviewers unconvinced that the authors thoroughly addressed related work, further weakening the paper's contributions. These issues, combined with scalability and experimental concerns, led to the decision to reject.

**Additional Comments On Reviewer Discussion:**

During the rebuttal period, reviewers raised key concerns about the scalability of the proposed mechanism, RealFM, as experiments were limited to 8 and 16 devices, far below real-world FL scenarios. While the authors argued that computational constraints caused this limitation and claimed RealFM could scale, this response was insufficient to address the scalability concern. Reviewers also questioned the “realistic” claims of the mechanism, highlighting ethical risks and fragile assumptions in heterogeneous settings, which the authors failed to fully alleviate. The literature review and comparisons with prior work, particularly Karimireddy et al., were deemed insufficient, with reviewers unconvinced by the proposed differentiations. Experimental inconsistencies, such as utility discrepancies across datasets, further weakened confidence in the results. Ultimately, the unresolved issues in scalability, ethical concerns, and empirical rigor outweighed the paper’s theoretical contributions, leading to the decision to reject.

---

### Decision · Program_Chairs · 2025-01-22

Reject